# Off-Policy Learning in Large Action Spaces: Optimization Matters More Than Estimation

Imad Aouali [* 1]   Otmane Sakhi [* 1]

## Abstract

Off-policy evaluation (OPE) and off-policy learning (OPL) are foundational for decision-making in offline contextual bandits. Recent advances in OPL primarily optimize OPE estimators with improved statistical properties, assuming that better estimators inherently yield superior policies. Although theoretically justified, this estimator-centric approach neglects a critical practical obstacle: challenging optimization landscapes. In this paper, we provide theoretical insights and empirical evidence showing that current OPL methods encounter severe optimization issues, particularly as the action space grows. We show that estimator-aware policy parametrization can mitigate, but not fully resolve, optimization challenges. Building on this, we explore simpler weighted log-likelihood objectives and demonstrate that they enjoy substantially better optimization properties and still recover competitive, often superior, learned policies. Our findings emphasize the necessity of explicitly addressing optimization considerations in the development of OPL algorithms for large action spaces.

## 1. Introduction

The offline contextual bandit (Dudík et al., 2011) leverages logged data from past interactions to improve future decision-making, with wide applications in areas like recommendation systems (Bottou et al., 2013; Aouali et al., 2022). We consider a standard setting where we are given a dataset $\mathcal{D}_n = \{(X_i, A_i, R_i)\}_{i=1}^n$ of $n$ i.i.d. tuples. Each tuple consists of a context $X_i \in \mathcal{X} \subseteq \mathbb{R}^d$ drawn from an unknown distribution $\nu$, an action $A_i \in \mathcal{A} = [K]$ sampled from a known logging policy as $A_i \sim \pi_0(\cdot \mid X_i)$, and a

---
*Equal contribution [1]Criteo AI Lab, Paris, France. Correspondence to: Imad Aouali <i.aouali@criteo.com>, Otmane Sakhi <o.sakhi@criteo.com>.

*Proceedings of the $43^{rd}$ International Conference on Machine Learning*, Seoul, South Korea. PMLR 306, 2026. Copyright 2026 by the author(s).

corresponding reward $R_i \sim p(\cdot \mid X_i, A_i)$ sampled from the unknown reward distribution $p(\cdot \mid X_i, A_i)$, whose mean is $r(x, a) = \mathbb{E}_{R \sim p(\cdot \mid x, a)}[R]$. The performance of any new policy $\pi$ is measured by its *value*, defined as the expected reward it would obtain:

$$V(\pi) = \mathbb{E}_{X \sim \nu, A \sim \pi(\cdot \mid X)}[r(X, A)]. \tag{1}$$

The goal of *off-policy learning (OPL)* is to leverage the offline dataset $\mathcal{D}_n$ to learn a policy $\hat{\pi}_n$ from a policy class $\Pi$ that maximizes this value, i.e., $\hat{\pi}_n = \arg\max_{\pi \in \Pi} V(\pi)$.

The dominant paradigm in OPL is to optimize an *off-policy evaluation (OPE)* estimator $\hat{V}_n(\pi)$ that approximates the true policy value $V(\pi)$ (Swaminathan & Joachims, 2015a) such as $\hat{V}_n(\pi) \approx V(\pi)$. This estimator is often based on the *inverse propensity scoring* (IPS) (Horvitz & Thompson, 1952) principle. The learning problem is thus framed as $\hat{\pi}_n = \arg\max_\pi \hat{V}_n(\pi)$, with the rationale that maximizing a more accurate estimate of the value yields a superior learned policy. However, this estimator-centric view overlooks a critical aspect: the optimization landscape. These IPS-based objectives (Dudík et al., 2011; Dudík et al., 2012; Dudik et al., 2014; Wang et al., 2017; Farajtabar et al., 2018; Su et al., 2020; Metelli et al., 2021; Kuzborskij et al., 2021; Saito & Joachims, 2022) are highly non-concave with common parameterized policies (Chen et al., 2019), prone to suboptimal local maxima, an issue more pronounced in large action spaces. Notably, even sophisticated estimators designed to reduce variance fail to overcome this optimization barrier, suffering from difficult-to-optimize landscapes.

We show that a practical way to mitigate these difficulties is through *estimator-aware policy parametrization*: choosing the policy class so that its structure matches the estimator's implicit biases. Concretely, this means restricting or factorizing the policy in ways that preserve the estimator's maximizers while reducing the degrees of freedom the optimizer must search over. This can substantially shorten optimization plateaus and improve robustness to hyperparameters. Importantly, this is a *compatibility* principle rather than a new estimator: it improves the trainability of existing IPS-based objectives without changing their statistical target. At the same time, it does not remove the fundamental non-concavity that arises when directly maximizing IPS-

based estimators with standard parametrizations, so difficult landscapes can persist in large action spaces.

Motivated by this, we study *policy-weighted log-likelihood (PWLL)* objectives, i.e., reward/advantage-weighted behavioral cloning updates that are widely used as policy-improvement objectives in offline RL and admit a direct contextual-bandit instantiation. We do not position PWLL as a novel method; rather, we use it as a representative family of *optimization-friendly* objectives whose log-likelihood form yields more benign, better-conditioned optimization dynamics than direct value maximization. PWLL is not used as an off-policy *value estimator* for evaluation or policy selection; its role is policy learning. Our theoretical and empirical results show that, in the large-action regimes we consider, prioritizing optimization stability can yield substantially better learned policies than directly maximizing sophisticated IPS-based estimators, even when the latter have stronger estimation guarantees.

To summarize, we make the following contributions.

1. **Optimization theory that scales with the action space**. We formalize how optimization pathologies, in particular long plateaus and exponentially many poor local maxima for IPS-based objectives, can scale with the number of actions $K$, explaining why these objectives get harder to optimize as $K$ increases.

2. **Objective-aware policy parametrization.** Using closed-form oracle policies (policies that maximize the objectives in expectation), we identify minimal sufficient policy structures for studied objectives and show how restricting or factorizing the learning policies accordingly can improve trainability and performance.

3. **PWLL as an optimization-friendly objective.** We study PWLL objectives and show that for linear-softmax policies the resulting objective is concave (strongly concave with $\ell_2$ regularization), yielding well-behaved optimization.

4. **Large-scale empirical stress tests.** On MovieLens (60k), Twitch (200k), and GoodReads (1M), we benchmark IPS-based and PWLL objectives under controlled optimization sweeps (batch size, learning-rate schedules, seeds), demonstrating that an objective's trainability dominates its estimation accuracy as a predictor of learned-policy quality in the large action space regimes we consider.

5. **Actionable guidance.** We distill practical design rules for large action spaces: when the goal is policy learning, use objective-aware policy parametrizations to improve trainability and prefer PWLL-style objectives to ensure better optimization stability.

Code and artifacts are publicly available here[1].

## 2. Analysis of IPS-Based Objectives

IPS-based objectives optimize an estimator $\hat{V}(\pi)$ of the policy value $V(\pi)$. To understand the policies to which these estimators converge, we study their *oracle policies* $\pi_*^{\mathrm{METHOD}} = \mathrm{argmax}_\pi \mathbb{E}[\hat{V}^{\mathrm{METHOD}}(\pi)]$. Taking the expectation removes sampling fluctuations and isolates the inductive bias of each objective: different estimators yield different oracle policies, even with infinite data. Crucially, oracle policies admit closed-form expressions, enabling precise characterization of each estimator's implicit bias. This analysis motivates *objective-aware parametrizations* that align the policy class with the estimator's bias to ease optimization: the first improvement we propose in this paper.

### 2.1. Standard IPS-Based Objectives

The foundational IPS estimator (Horvitz & Thompson, 1952) re-weights observed rewards by the ratio between the target policy $\pi$ and the logging policy $\pi_0$:

$$\hat{V}_{\mathrm{IPS}}(\pi) = \frac{1}{n} \sum_{i=1}^{n} \frac{\pi(A_i \mid X_i)}{\pi_0(A_i \mid X_i)} R_i. \qquad (2)$$

In expectation, IPS selects the best-rewarding action among those in the support of $\pi_0$:

$$\pi_*^{\mathrm{IPS}}(a \mid x) = \mathbb{1}\left[a = \underset{a' \in \mathcal{A}}{\mathrm{argmax}}\, r(x, a') \mathbb{1}[\pi_0(a' \mid x) > 0]\right].$$

**Clipped IPS (cIPS).** To mitigate the high variance of IPS, a widely used variant is cIPS (Bottou et al., 2013) that clips small propensity scores at a threshold $\tau \in (0, 1)$:

$$\hat{V}_{\mathrm{CIPS}}(\pi) = \frac{1}{n} \sum_{i=1}^{n} \frac{\pi(A_i \mid X_i)}{\max\{\pi_0(A_i \mid X_i), \tau\}} R_i. \qquad (3)$$

This clipping introduces a bias. The oracle policy downweights the rewards of rare actions, causing it to favor actions that were frequent under $\pi_0$, even if they are suboptimal:

$$\pi_*^{\mathrm{CIPS}}(a \mid x) = \mathbb{1}\left[a = \underset{a' \in \mathcal{A}}{\mathrm{argmax}}\, \frac{\pi_0(a' \mid x) r(x, a')}{\max\{\pi_0(a' \mid x), \tau\}}\right].$$

**Exponential smoothing (ES).** Instead of hard clipping, ES (Aouali et al., 2023) smooths importance weights by raising propensities to a fractional power $\alpha \in (0, 1)$:

$$\hat{V}_{\mathrm{ES}}(\pi) = \frac{1}{n} \sum_{i=1}^{n} \frac{\pi(A_i \mid X_i)}{\pi_0(A_i \mid X_i)^\alpha} R_i. \qquad (4)$$

---

[1] https://github.com/imadaouali/opl-las-pwll

Its oracle policy balances reward maximization with preference for frequent actions:

$$\pi_*^{\text{ES}}(a \mid x) = \mathbb{1}\left[a = \operatorname*{argmax}_{a' \in \mathcal{A}} r(x, a')\pi_0(a' \mid x)^{1-\alpha}\right].$$

Another variant of ES regularizes the entire importance weight as $\left(\frac{\pi}{\pi_0}\right)^\beta$ instead of only the denominator. In contrast to the deterministic policies derived from IPS, cIPS, and the ES formulation above, this approach yields a stochastic oracle policy: $\pi_*^{\text{ES}}(a \mid x) \propto r(x, a)^{1/(1-\beta)}\pi_0(a \mid x)$. Other regularizations include logarithmic smoothing (Sakhi et al., 2024), implicit exploration (Gabbianelli et al., 2024), harmonic correction (Metelli et al., 2021), shrinkage (Su et al., 2020). We focus on ES and cIPS which we deem to be good representatives already.

**Doubly robust (DR).** The DR estimator incorporates a reward model $\hat{r}(x, a)$ to reduce variance and enable generalization to actions outside $\pi_0$'s support. A common clipped variant is:

$$\hat{V}_{\text{DR}}(\pi) = \frac{1}{n}\sum_{i=1}^{n}\frac{\pi(A_i \mid X_i)}{\max\{\pi_0(A_i \mid X_i), \tau\}}\left(R_i - \hat{r}(X_i, A_i)\right)$$
$$+ \mathbb{E}_{A \sim \pi(\cdot \mid X_i)}\left[\hat{r}(X_i, A)\right]. \quad (5)$$

Its oracle policy interpolates between the reward model prediction and an importance weighting correction for the reward model error:

$$\pi_*^{\text{DR}}(a \mid x) = \mathbb{1}\Big[a = \operatorname*{argmax}_{a' \in \mathcal{A}} \hat{r}(x, a')$$
$$+ \frac{\pi_0(a' \mid x)}{\max\{\pi_0(a' \mid x), \tau\}}\left(r(x, a') - \hat{r}(x, a')\right)\Big].$$

### 2.2. Large-Scale IPS-Based Objectives

In large action spaces, importance weights $\frac{\pi(a \mid x)}{\pi_0(a \mid x)}$ can become huge, leading to estimators with high variance. To mitigate this, modern methods compute marginalized importance weights over a lower-dimensional action representation, trading bias for reduced variance.

**Marginalized IPS (MIPS).** MIPS (Saito & Joachims, 2022) tackles large action spaces by clustering actions. It maps each action $a$ to a cluster $c$ via a function $h : \mathcal{A} \to \mathcal{C}$, where $|\mathcal{C}| \ll |\mathcal{A}|$. Estimation is then performed at the cluster level:

$$\hat{V}_{\text{MIPS}}(\pi) = \frac{1}{n}\sum_{i=1}^{n}\frac{\pi(C_i \mid X_i)}{\pi_0(C_i \mid X_i)}R_i, \quad (6)$$

where $C_i = h(A_i)$ and $\pi(c \mid x) = \sum_{a \in c}\pi(a \mid x)$.

This cluster-level marginalization introduces bias: the oracle policy only selects the best *cluster* based on its average reward under $\pi_0$, and cannot differentiate between actions within that cluster:

$$\pi_*^{\text{MIPS}}(c \mid x) = \mathbb{1}\Big[c = \operatorname*{argmax}_{c' \in \mathcal{C}}\Big\{\frac{\sum_{a \in c'}\pi_0(a \mid x)r(x, a)}{\sum_{a \in c'}\pi_0(a \mid x)}\Big\}\Big].$$

Hence, MIPS offers no specific guidance for selecting an action within the optimal cluster; any action is considered equally valid. Consequently, one possible induced action-level oracle under uniform tie-breaking is:

$$\pi_*^{\text{MIPS}}(a \mid x)$$
$$= \frac{\mathbb{1}\Big[h(a) = \operatorname{argmax}_{c' \in \mathcal{C}}\Big\{\frac{\sum_{a \in c'}\pi_0(a \mid x)r(x, a)}{\sum_{a \in c'}\pi_0(a \mid x)}\Big\}\Big]}{|h(a)|}.$$

where $|h(a)|$ denotes the size of the cluster containing $a$.

**Conjunct effect modeling (OffCEM).** Building on MIPS, OffCEM (Saito et al., 2023) uses a reward model $\hat{r}$ to correct for the cluster-level aggregation bias, in a doubly robust fashion:

$$\hat{V}_{\text{OFFCEM}}(\pi) = \frac{1}{n}\sum_{i=1}^{n}\frac{\pi(C_i \mid X_i)}{\pi_0(C_i \mid X_i)}\left(R_i - \hat{r}(X_i, A_i)\right)$$
$$+ \mathbb{E}_{A \sim \pi(\cdot \mid X_i)}[\hat{r}(X_i, A)]. \quad (7)$$

The resulting oracle policy selects the action that maximizes the model-predicted reward $\hat{r}$, plus a cluster-level correction term that accounts for model error:

$$\pi_*^{\text{OffCEM}}(a \mid x) = \mathbb{1}\Big[a = \operatorname*{argmax}_{a' \in \mathcal{A}}\Big\{\hat{r}(x, a')$$
$$+ \frac{\sum_{\bar{a} \in h(a')}\pi_0(\bar{a} \mid x)(r(x, \bar{a}) - \hat{r}(x, \bar{a}))}{\sum_{\bar{a} \in h(a')}\pi_0(\bar{a} \mid x)}\Big\}\Big].$$

**Two-stage decomposition (POTEC).** In this paper, we argue that POTEC (Saito et al., 2025) should be viewed as an *optimization strategy* of OffCEM (rather than a new estimator). It restricts the policy to a cluster-informed form,

$$\pi(a \mid x) = \sum_{c \in \mathcal{C}}\pi^{\text{RM}}(a \mid x, c)\pi^{\text{CL}}(c \mid x),$$

where $\pi^{\text{RM}}(a \mid x, c) = \mathbb{1}[a = \operatorname{argmax}_{a' \in c}\hat{r}(x, a')]$ is a fixed, model-based policy that deterministically selects the best action within each cluster. Learning is then simplified to finding the optimal cluster-level policy $\pi^{\text{CL}}$ that maximizes the OffCEM objective in Equation (7):

$$\hat{V}_{\text{POTEC}}(\pi^{\text{CL}}) = \frac{1}{n}\sum_{i=1}^{n}\frac{\pi^{\text{CL}}(C_i \mid X_i)}{\pi_0(C_i \mid X_i)}\left(R_i - \hat{r}(X_i, A_i)\right)$$
$$+ \sum_{c \in \mathcal{C}}\pi^{\text{CL}}(c \mid X_i)\hat{r}_c^*(X_i), \quad (8)$$

where $\hat{r}_c^*(x) = \max_{a \in c} \hat{r}(x, a)$ is the estimated reward of the best action in cluster $c$. This practical decomposition has the same optimal oracle policy as `OffCEM`:

$$\pi_*^{\text{POTEC}} = \pi_*^{\text{OffCEM}}.$$

**Policy convolution (`PC`).** Moving beyond hard clustering, `PC` (Sachdeva et al., 2023) leverages the assumption that actions close in an embedding space yield similar rewards. For each action $a$, it aggregates over its neighborhood of nearest neighbors $N_\epsilon(a) = \{a' : d(a, a') < \epsilon\}$, where $d$ is a pre-defined distance metric (e.g., $\ell_2$ distance between action embeddings):

$$\hat{V}_{\text{PC}}(\pi) = \frac{1}{n} \sum_{i=1}^n \frac{\pi(N_\epsilon(A_i) \mid X_i)}{\pi_0(N_\epsilon(A_i) \mid X_i)} R_i, \qquad (9)$$

$$\text{with } \pi(N_\epsilon(a) \mid x) = \sum_{a' \in N_\epsilon(a)} \pi(a' \mid x).$$

The induced oracle policy is deterministic: it selects the action $a'$ that maximizes an aggregated neighborhood score. Each logged neighbor $\bar{a} \in N_\epsilon(a')$ contributes its reward $r(x, \bar{a})$, weighted by the conditional probability of observing $\bar{a}$ under the logging policy restricted to its neighborhood.

$$\pi_*^{\text{PC}}(a \mid x) = \mathbb{I}\Big[a = \underset{a' \in \mathcal{A}}{\operatorname{argmax}} \Big\{ \sum_{\bar{a} \in N_\epsilon(a')} \frac{\pi_0(\bar{a} \mid x) r(x, \bar{a})}{\pi_0(N_\epsilon(\bar{a}) \mid x)} \Big\} \Big].$$

Other recent IPS variants for large action spaces (Peng et al., 2023; Cief et al., 2024; Taufiq et al., 2024) are often extensions of MIPS that relax its core assumptions. We focused on four methods (`MIPS`, `OffCEM`, `POTEC`, and `PC`), which we consider representative of this family. Since these variants largely share the same MIPS foundation and optimization procedure (with the notable exception of `POTEC`), we expect our findings to be generally applicable.

### 2.3. Optimization Challenges

The effectiveness of IPS-based estimators in off-policy learning is often limited by their challenging optimization landscape. These objectives become difficult to optimize when paired with standard, expressive policy classes such as the softmax. This section explores why this occurs and introduces *objective-aware parametrization* as a strategy to mitigate, though not entirely solve, the problem.

To analyze the optimization process, we consider policies parametrized by a softmax function over an *effective action space*[2] $\mathcal{A}_{\text{eff}} \subseteq \mathcal{A}$, which is the set of actions that can be assigned non-zero probability. By default, $\mathcal{A}_{\text{eff}} = \mathcal{A}$, but we explain below why restricting it to match the structure of

---

[2]The effective action space can also depend on context $x$, i.e., $\mathcal{A}_{\text{eff}}(x) \subseteq \mathcal{A}$. We omit this dependence for notational simplicity.

the estimator's oracle policy can be beneficial. Specifically, the policy takes the form for all $a \in \mathcal{A}$:

$$\pi_\theta(a \mid x) = \frac{\exp(s_\theta(x, a))}{\sum_{a' \in \mathcal{A}_{\text{eff}}} \exp(s_\theta(x, a'))} \mathbb{1}_{a \in \mathcal{A}_{\text{eff}}}, \qquad (10)$$

where $s_\theta(x, a)$ is a learnable score function. Common choices are linear softmax scores:

$$\text{lightweight: } s_\theta(x, a) = \phi(x, a)^\top \theta,$$

$$\text{heavyweight: } s_\theta(x, a) = \phi(x)^\top \theta_a, \qquad (11)$$

which we call *lightweight parametrization* (a single shared parameter vector $\theta$, corresponding to a joint reward model) and *heavyweight parametrization* (separate parameters $\theta_a$ for each action, corresponding to a disjoint reward model).

The size of the effective action space, $K_{\text{eff}} = |\mathcal{A}_{\text{eff}}|$, is the critical factor governing optimization difficulty. The following propositions (proofs in Appendix C, adapted from Chen et al. (2019); Mei et al. (2020a)) reveal the severity of the problem.

First, gradient-based methods can become trapped in suboptimal regions for extended periods.

**Proposition 2.1** (Optimization plateaus). *For any IPS-based estimator $\hat{V}$ that is linear in $\pi$, even with a linear softmax policy, there exist problem instances where gradient ascent remains trapped in a suboptimal region for $\mathcal{O}(K_{\text{eff}})$ iterations.*

Second, the optimization landscape has numerous poor local maxima.

**Proposition 2.2** (Local maxima). *Under similar conditions, the optimization landscape for IPS-based objectives can contain a number of local maxima that is exponential in $K_{\text{eff}}$.*

These results highlight that $K_{\text{eff}}$ plays a central role in optimization difficulty. The standard choice of $\mathcal{A}_{\text{eff}} = \mathcal{A}$, which sets $K_{\text{eff}} = K$, leads to optimization failure in large action spaces where $K$ can reach millions: learning must navigate a landscape with potentially $\mathcal{O}(K)$-length plateaus and exponentially many local maxima.

Surprisingly, even sophisticated methods designed specifically for large action spaces often fall into this trap. At first glance, methods such as `MIPS`, `OffCEM`, and `PC` appear to operate in a smaller space because their objectives involve marginalized probabilities: $\pi(C_i \mid X_i)$ in `MIPS` and `OffCEM`, or $\pi(N_\epsilon(A_i) \mid X_i)$ in `PC`. However, these marginalized terms are defined as sums over an underlying

action-level policy:

$$\pi(C_i \mid X_i) = \sum_{a \in C_i} \pi(a \mid X_i),$$

$$\pi(N_\epsilon(a) \mid x) = \sum_{a' \in N_\epsilon(a)} \pi(a' \mid x).$$

Then, if $\pi(a \mid x)$ is a softmax over $\mathcal{A}$, then $K_{\text{eff}} = K$ and Propositions 2.1 and 2.2 apply with $K_{\text{eff}} = K$ which is large. The only exception is POTEC, which fixes the intra-cluster policy $\pi^{\text{RM}}$ and only optimizes a cluster-level policy $\pi^{\text{CL}}$. This reduces the effective action space to $\mathcal{A}_{\text{eff}} = \mathcal{C}$ with $K_{\text{eff}} = |\mathcal{C}| \ll K$, directly mitigating the optimization pathologies.

## 2.4. Design Implications: Objective-Aware Parametrization

The choice of $K_{\text{eff}}$ introduces a fundamental trade-off. A smaller effective action space simplifies the optimization landscape, but risks excluding the optimal action and reduces policy expressiveness. If $\mathcal{A}_{\text{eff}}$ is chosen arbitrarily, it may degrade performance. The challenge is to find the *sweet spot*: a parametrization constrained enough to be optimizable, yet expressive enough to contain the objective's maximizer.

This is precisely where our asymptotic analysis helps. The oracle policy $\pi_*^{\text{METHOD}}$ reveals the minimal sufficient set of actions required to maximize each objective. By aligning the policy parametrization with this structure, we can reduce $K_{\text{eff}}$ without sacrificing performance: the core principle of our proposed *objective-aware parametrization*.

For instance, the oracle policies for IPS, cIPS, and ES are confined to the support of the logging policy, $S_0(x)$. This implies that $\mathcal{A}_{\text{eff}} = S_0$ is sufficient, reducing $K_{\text{eff}}$ from $K$ to $|S_0| \ll K$. Similarly, for OffCEM and MIPS, the cluster-level structure of their oracle policies suggests a two-stage decomposition similar to that of POTEC, reducing $K_{\text{eff}}$ to $|\mathcal{C}|$. We summarize these observations as claims, validated empirically in Section 4:

**Claim 2.3.** *For IPS, cIPS, and ES, restricting the policy support to $S_0$ reduces $K_{\text{eff}}$ and yields superior learned policies.*

**Claim 2.4.** *For OffCEM and MIPS, a two-stage POTEC-style decomposition that optimizes at the cluster level outperforms action-level parametrization.*

While objective-aware parametrization mitigates the optimization pathologies of Propositions 2.1 and 2.2 by reducing $K_{\text{eff}}$, it only treats the symptoms without curing the underlying non-concavity. In the next section, we propose a more fundamental shift: abandoning value estimation in favor of inherently tractable objectives.

## 3. Analysis of PWLL objectives

To overcome the optimization challenges of IPS-based objectives, we consider policy-weighted log-likelihood (PWLL) objectives. These methods trade accurate value estimation for a well-behaved, concave optimization landscape, leading to more robust and effective policy learning.

**General form.** Given a positive weighting function $g(r, p_0)$, the PWLL objective is:

$$\hat{U}_g(\pi) = \frac{1}{n} \sum_{i=1}^{n} g(R_i, \pi_0(A_i \mid X_i)) \log \pi(A_i \mid X_i). \quad (12)$$

The key motivation behind PWLL is to replace the linear dependence on the policy in IPS-based estimators, responsible for plateaus and local maxima in Section 2, with a concave transformation. Softmax policies are parametrized through scores $s_\theta(x, a)$, and the map $s \mapsto \log \text{softmax}(s)$ is concave. Consequently, for common linear parametrizations in Equation (11), the composition $\log \pi_\theta(a \mid x)$ is concave in $\theta$. This removes the optimization pathologies inherent to IPS-based objectives. Proposition 3.1 (proof in Appendix C) formalizes this advantage.

**Proposition 3.1.** *For linear softmax policies $\pi_\theta$, the PWLL objective $\hat{U}_g(\pi_\theta)$ is concave in $\theta$. With $\ell_2$ regularization, it is strongly concave.*

Proposition 3.1 makes PWLL appealing for stochastic optimization. In Appendix D, we show that under standard assumptions of bounded feature norms $\|\phi(x, a)\|$ and weights $g(R_i, \pi_0(A_i \mid X_i))$, these objectives satisfy the regularity conditions necessary to invoke established convergence theorems (Garrigos & Gower, 2023). This allows us to derive problem-dependent convergence guarantees: stochastic gradient ascent attains a global $\mathcal{O}(1/\sqrt{T})$ rate in the general (concave) case (Proposition D.4), accelerating to a geometric rate under $\ell_2$-regularization (Proposition D.5).

Beyond optimization properties, PWLL also admits a simple statistical interpretation. $\hat{U}_g(\pi)$ in Equation (12) is a weighted log-likelihood: the term $\log \pi(A_i \mid X_i)$ performs standard behavior cloning, while the weight $g(R_i, \pi_0(A_i \mid X_i))$ determines how *desirable*[3] each logged sample is. This turns off-policy learning into a form of logging-aware and reward-weighted maximum-likelihood estimation. Different choices of $g$ encode different notions of desirability. For example, the weighting

$$g(r, \pi_0(a \mid x)) = \frac{r}{\max\{\pi_0(a \mid x), \tau\}}$$

emphasizes samples with high reward while reducing the influence of actions that the logging policy selected very

---

[3]By how desirable an action is, we mean how strongly this action should influence the learned policy.

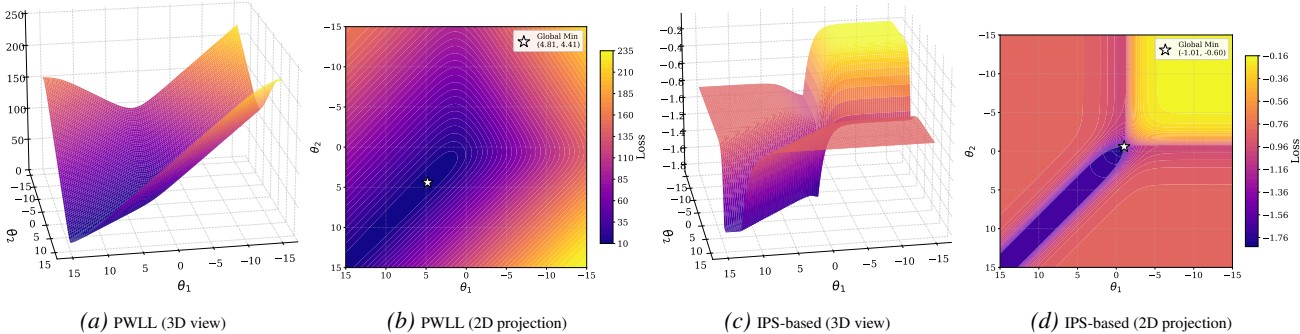

*(a)* PWLL (3D view)  *(b)* PWLL (2D projection)  *(c)* IPS-based (3D view)  *(d)* IPS-based (2D projection)

*Figure 1.* Optimization landscapes on a toy example. PWLL (`cLPI`) vs IPS-based (`cIPS`).

frequently. At the same time, the clipping at $\tau$ prevents extremely rare actions from receiving disproportionately large weights, ensuring that their contribution is attenuated once $\pi_0(a \mid x)$ falls below the threshold. In this view, desirable samples are those that provide strong reward evidence without allowing very small propensities to dominate the updates. Many other PWLL variants arise from different choices of $g$ (see below), each specifying a distinct prioritization scheme for the logged data, while all benefit from the concavity induced by the logarithmic term.

To illustrate the qualitative difference between PWLL and IPS-based objectives, we construct a simple offline bandit problem with $K = 3$ actions and visualize the resulting optimization landscapes in a two-parameter policy space. Concretely, we consider a non-contextual setting with deterministic mean rewards $r = (0.9, 0.7, 0.2)$ and a logging policy $\pi_0$ whose support places almost all mass on action 3 ($\pi_0(1) = 0.002$, $\pi_0(2) = 0.003$, $\pi_0(3) = 0.995$). We generate a fixed dataset of $n = 60$ logged samples $(A_i, R_i)$ by drawing actions $A_i \sim \pi_0$ and binary rewards from the corresponding Bernoulli distributions, $R_i \sim \text{Bern}(r(A_i))$. To obtain a two-dimensional visualization, we parameterize the target policy using a softmax over three logits: $\pi_\theta(a) = e^{\theta_a} / \sum_{b \in [3]} e^{\theta_b}$, fixing the logit associated with action 3 as $\theta_3 = 1$, and letting the remaining two logits be free parameters $(\theta_1, \theta_2)$.

In Figure 1, the PWLL landscape is concave with well-scaled gradients, and optimization trajectories converge reliably from roughly any initialization. In contrast, the IPS-based landscape consists of flat regions, separated by a narrow band of extremely steep curvature. This creates both vanishing and exploding gradients, severe ill-conditioning, and high sensitivity to initialization and learning rate. This aligns with the optimization pathologies in Propositions 2.1 and 2.2.

*Remark* 3.2 (Beyond linear-softmax policies). The concavity guarantee of Proposition 3.1 assumes linear-softmax policies. In many large-scale recommendation systems, a deep encoder is pre-trained and kept fixed, and only a final linear

head is optimized for the downstream task; in this case, the policy is still linear-softmax in the trainable parameters, and PWLL objectives retain their concavity. When the full network is trained end-to-end, concavity no longer holds. Yet, PWLL's gradients $g(R_i, \pi_0(A_i|X_i)) \nabla_\theta \log \pi_\theta(A_i \mid X_i)$ match the structure of cross-entropy gradients, which are known to produce stable and well-scaled updates in deep architectures. Thus, even without formal guarantees, PWLL maintains substantially more benign optimization dynamics than IPS-based objectives.

**Local policy improvement (`LPI`).** Liang & Vlassis (2022) set $g(r, p_0) = r$, which optimizes the log-likelihood of actions weighted by their observed rewards:

$$\hat{U}_{\text{LPI}}(\pi) = \frac{1}{n} \sum_{i=1}^{n} R_i \log \pi(A_i \mid X_i). \tag{13}$$

The oracle policy balances reward-seeking with imitation of the logging policy:

$$\pi_*^{\text{LPI}}(a \mid x) \propto r(x, a) \pi_0(a \mid x).$$

**Clipped LPI (`cLPI`)** uses importance-weight clipping, setting $g(r, p_0) = \frac{r}{\max(p_0, \tau)}$:

$$\hat{U}_{\text{CLPI}}(\pi) = \frac{1}{n} \sum_{i=1}^{n} \frac{R_i}{\max\{\pi_0(A_i \mid X_i), \tau\}} \log \pi(A_i \mid X_i). \tag{14}$$

In a similar spirit to `cIPS`, its oracle policy corrects for action frequency under $\pi_0$, down-weighting the influence of rare actions due to the clipping:

$$\pi_*^{\text{CLPI}}(a \mid x) \propto r(x, a) \frac{\pi_0(a \mid x)}{\max\{\pi_0(a \mid x), \tau\}}.$$

An immediate consequence is that `cLPI` and `cIPS` induce the same greedy oracle decision rule. Although $\pi_*^{\text{CLPI}}$ is generally stochastic, it is proportional to the same clipped

reward score that defines the deterministic `cIPS` oracle. Therefore:

$$\pi_*^{\text{cIPS}}(a \mid x) = \mathbb{1}\left[a = \arg\max_{a' \in \mathcal{A}} \pi_*^{\text{cLPI}}(a' \mid x)\right].$$

Thus, if the goal is to recover the `cIPS` oracle decision rule, one can instead optimize the `cLPI` objective and deploy the greedy policy obtained by taking the argmax of the learned probabilities. This provides an optimization-friendly workaround for targeting the `cIPS` oracle without directly optimizing the non-concave `cIPS` objective. Our experiments in the next section support this strategy, showing that greedy policies learned with `cLPI` consistently outperform their `cIPS` counterparts.

**KL regularization (`RegKL`).** To further amplify the reward signal relative to the logging policy prior, RegKL uses an exponential weighting function $g(r, p_0) = \exp(r/\beta)$:

$$\hat{U}_{\text{REGKL}}(\pi) = \frac{1}{n} \sum_{i=1}^{n} \exp(R_i/\beta) \log \pi(A_i \mid X_i). \quad (15)$$

The oracle policy is proportional to the logging policy, weighted by the exponentiated reward:

$$\pi_*^{\text{RegKL}}(a \mid x) \propto \mathbb{E}_{r \sim p(\cdot \mid x,a)}\left[\exp(r/\beta)\right] \pi_0(a \mid x).$$

The temperature parameter $\beta$ smoothly interpolates between behavior cloning ($\beta \to \infty$) and greedy reward maximization ($\beta \to 0$).

Note that BPR (Rendle et al., 2012) can be seen as an approximate PWLL objective, and we included it in our experiments. In fact, this general form of PWLL lends itself to numerous variations by modifying the weighting function $g$. For instance, one could introduce variants inspired by regularized IPS like exponential smoothing. While many such variants can be proposed for specific use cases, the central message of our work is that the well-behaved optimization landscape of the PWLL family is of greater practical importance than the estimation accuracy of IPS-based objectives. Thus, an exploration of these PWLL variants is beyond our scope. We contend that the foundational methods analyzed above, LPI, cLPI, and RegKL, along with the widely used BPR are sufficient to demonstrate the inherent advantages of PWLL objectives.

**Positioning w.r.t. offline RL and contextual bandits.** The PWLL objectives we study are not introduced as fundamentally new policy-update rules. They can be viewed as the one-step contextual-bandit specialization of reward/advantage-weighted behavioral cloning updates used in offline RL (Nair et al., 2020; Wang et al., 2020; Peng et al., 2019; Peters, 2006). The difference is that in our

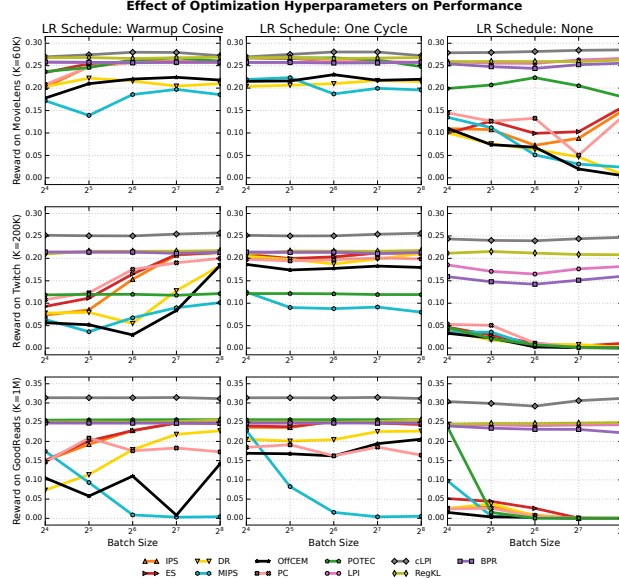

*Figure 2.* Effect of batch size and learning rate schedule on final validation reward using three large-scale datasets. IPS-based objectives are highly sensitive, while PWLL objectives are robust.

bandit setting, the weights $g(R_i, \pi_0(A_i \mid X_i))$ can be computed directly from logged rewards and known propensities, without value-function estimation or bootstrapping.

Importantly, although behavior-cloning-style objectives are common in offline RL, the contextual bandit framework remains a dominant choice in large-scale recommender systems due to its simplicity and ease of deployment. In that literature, OPL is much more often framed as maximizing IPS-based objectives, while (reward/advantage-)weighted cloning objectives are comparatively less common. This gap in practice motivates our focus on trainability for large-$K$ contextual-bandit OPL.

Our contribution is therefore not necessarily algorithmic novelty, but a large-action contextual-bandit diagnosis and a set of design rules: (i) we characterize how IPS-based objectives become hard to optimize as the effective action set size $K_{\text{eff}}$ grows, (ii) we derive *objective-aware parametrizations* from oracle policies to reduce $K_{\text{eff}}$ without changing the objective's asymptotic target, and (iii) we provide large-scale evidence (60k-1M actions) via controlled optimization stress tests.

## 4. Empirical Analysis

We conduct our empirical evaluation on three large-scale recommendation datasets: MovieLens ($K = 60$k) (Lam & Herlocker, 2016), Twitch ($K = 200$k) (Rappaz et al., 2021), and GoodReads ($K = 1$M) (Wan et al., 2019). These benchmarks feature action spaces with up to one million items, representing some of the largest settings studied

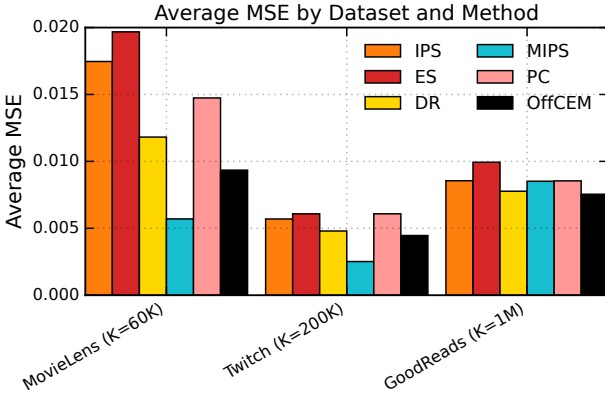

*Figure 3.* Average MSE by Dataset and Method. PWLL methods are excluded as their very high MSE values would distort the scale and obscure the comparison.

in the offline policy learning literature. For all experiments, we employ the common softmax inner-product policies. We compare methods from both objective families. For IPS-based objectives, we include `IPS`, `ES`, `DR`, `MIPS`, `OffCEM`, `POTEC`, and `PC` in Section 2. For PWLL objectives, we evaluate `LPI`, `cLPI`, `RegKL`, and `BPR` in Section 3. All implementation details are provided in Appendix E.

### 4.1. Optimization is the Main Bottleneck

To test our central hypothesis that *optimization challenges are a more significant barrier than estimation accuracy*, we evaluate how objectives perform under various optimization configurations. If an algorithm's success is highly dependent on specific hyperparameters like batch size or learning rate, it suggests a difficult, non-robust optimization landscape. This experiment directly probes the practical trainability of each method, a key aspect our paper argues is often overlooked.

The results strongly support our claim. As shown in Figure 2, *IPS-based objectives are highly sensitive* to batch size and learning rate schedule: minor changes can cause performance collapse, making them difficult to tune and train reliably. In contrast, *PWLL objectives remain robust*, achieving consistently high reward across all configurations. This stability translates directly into better learned policies: *PWLL objectives outperform IPS-based objectives on all datasets*. Even `POTEC`, a state-of-the-art method designed for large action spaces, is surpassed by the much simpler and easier-to-optimize `cLPI`.

The figure also supports Claim 2.4. Indeed, there is a consistent performance gap between `POTEC` and `OffCEM`. Both methods are designed to maximize the same asymptotic objective as we show in Section 2; their statistical goals are identical. The divergence in performance, therefore, can be attributed entirely to their differing optimization strate-

gies. `POTEC`'s use of a two-stage, cluster-level optimization proves far more effective than `OffCEM`'s naive, action-level parametrization.

Finally, one might expect an objective designed for estimation fidelity, such as a low-MSE IPS-based estimator, to naturally induce a better learned policy. Our results show that this intuition is misleading. PWLL objectives are poor value estimators by design, with MSE values too large to display on the scale of Figure 3, yet they achieve the strongest policy-learning performance. Moreover, even within IPS-based methods, lower estimation error does not predict better learned policies: `ES` has consistently larger MSE than `MIPS` in Figure 3, but attains higher validation reward (Figure 2). These observations provide evidence against an estimator-centric view of OPL. In large action spaces, a stable optimization landscape is more critical for policy learning than statistical accuracy as a value estimator. For completeness, we also report the evolution of the methods' MSE during training in Appendix E.

### 4.2. Objective-Aware Parametrization

To empirically validate Claim 2.3, we compare a naive, whole-action-space parametrization against our proposed objective-aware approach, which restricts the policy's effective action space to the logging policy support, $S_0$. As shown for the `IPS` objective in Figure 4, the naive approach is highly unstable, with performance collapsing under simple learning configurations. In contrast, the objective-aware version is very robust, achieving high reward consistently across all batch sizes and schedules. This benefit extends even to inherently stable PWLL objectives like `cLPI`, which achieve even better performance with the restricted support. This provides strong evidence for Claim 2.3: aligning the policy structure with the objective's inductive bias simplifies the optimization landscape, leading to greater stability and superior learned policies. This finding holds across all datasets, with full results available in Appendix E.

**Additional ablations (Appendix E).** We further validate these conclusions via experiments with varying random seeds, reward-noise stress tests, hyperparameter sweeps, smaller action spaces ($K$), cluster/support sensitivity, pessimism ablations, etc.; see Appendix E for details.

## 5. Conclusion

The dominant approach to off-policy learning focuses on developing sophisticated IPS-based estimators while neglecting a crucial factor: the optimization landscape. We demonstrated, both theoretically and empirically, that this landscape becomes prohibitively difficult to optimize in large action spaces, undermining the practical effectiveness of even state-of-the-art estimators.

**Effect of Objective-Aware Parametrization on Performance**

*Figure 4.* The effect of objective-aware parametrization for `IPS` and `cLPI` on `MovieLens`.

Our analysis motivates two strategies. First, objective-aware policy parametrizations align the policy class with the estimator's inductive bias, reducing the effective search space. Second, PWLL objectives abandon value estimation entirely in favor of inherently concave optimization landscapes. Experiments confirm that this focus on optimization tractability yields more robust learning, reduced sensitivity to hyperparameters, and superior policies.

Our work has several limitations. First, PWLL objectives are not value estimators: they cannot be used for off-policy evaluation or policy selection (choosing the best policy from a finite candidate set) when accurate value estimates and their comparison are required. Second, the concavity guarantee of Proposition 3.1 holds only for linear-softmax policies; when training deep networks end-to-end, PWLL retains favorable gradient structure but loses formal concavity guarantees, although IPS-based objectives face even more severe optimization challenges in this setting. Third, PWLL's oracle policies inherently depend on the logging policy (e.g., $\pi_*^{\mathrm{LPI}} \propto r(x, a)\pi_0(a \mid x)$), which may be suboptimal when $\pi_0$ has poor coverage of high-reward actions; however, this limitation is shared by IPS-based methods, whose oracle policies similarly depend on $\pi_0$'s support.

## Impact Statement

This paper presents work whose goal is to advance the field of machine learning. There are many potential societal consequences of our work, none of which we feel must be specifically highlighted here.

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

# A. Extended related work

**Offline contextual bandits.** The contextual bandit framework is widely used for online learning under uncertainty (Lattimore & Szepesvari, 2019). Yet, many applications pose challenges for online exploration, motivating offline approaches that optimize decisions from logged data (Bottou et al., 2013). Since large datasets of past interactions are often available, policies can be improved without new experimentation (Swaminathan & Joachims, 2015a). This setting, known as offline (or off-policy) contextual bandits (Dudík et al., 2011), relies on off-policy evaluation (OPE) to estimate policy performance from logged data. These estimators are then used to learn value-maximizing policies (Off-policy learning, OPL).

**Off-policy evaluation.** OPE (Dudík et al., 2011; Dudík et al., 2012; Dudik et al., 2014; Wang et al., 2017; Farajtabar et al., 2018; Su et al., 2020; Metelli et al., 2021; Kuzborskij et al., 2021; Saito & Joachims, 2022; Sakhi et al., 2020; Jeunen & Goethals, 2021) has attracted significant attention in recent years, with methods falling into three main categories. The direct method (DM) fits a model to predict expected costs for each context–action pair and then uses it to estimate policy value (Jeunen & Goethals, 2021; Aouali et al., 2025), a strategy that has proven particularly effective in large-scale recommender systems (Sakhi et al., 2020; Jeunen & Goethals, 2021). Inverse propensity scoring (IPS) instead reweights observed outcomes to correct for the bias of the logging policy (Horvitz & Thompson, 1952; Dudík et al., 2012). While IPS is unbiased under absolute continuity, it is highly sensitive to variance and bias when this condition is violated (Sachdeva et al., 2020). A wide range of techniques has been proposed to address this issue, including clipping (Ionides, 2008; Bottou et al., 2013), shrinkage (Su et al., 2020), smoothing (Metelli et al., 2021; Aouali et al., 2023; Sakhi et al., 2024), implicit exploration (Gabbianelli et al., 2024), and self-normalization (Swaminathan & Joachims, 2015b), among others (Aouali et al., 2024). A third line of work combines these two approaches in doubly robust (DR) estimators, which integrate modeling with reweighting for improved bias–variance trade-offs (Robins & Rotnitzky, 1995; Dudík et al., 2011; Dudik et al., 2014; Farajtabar et al., 2018). Our work focuses on off-policy learning using these estimators.

**Off-policy learning.** OPL is typically built on DM, IPS, or DR. DM selects actions by maximizing predicted reward, either deterministically or stochastically, while IPS and DR optimize a parameterized policy via stochastic gradient descent (Swaminathan & Joachims, 2015a), where the unknown gradient of the true risk must be estimated using reweighting. Beyond these approaches, statistical learning tools have introduced new objectives grounded in PAC-based pessimism, providing stronger theoretical guarantees (London & Sandler, 2019; Sakhi et al., 2023a). Our contribution complements this literature by examining the optimization landscape of OPL in large action spaces, which remains largely underexplored.

**Large action spaces.** Regularization can improve IPS in moderate settings, but scaling to large action spaces requires additional structure. One prominent direction leverages action embeddings: for example, marginalized IPS (MIPS) (Saito & Joachims, 2022) reduces variance by exploiting embedding information while remaining unbiased if the embeddings capture the causal effects of actions on costs. High-dimensional embeddings, however, can still induce variance, and misspecified embeddings can introduce bias. Recent work addresses these issues by learning embeddings directly from data (Peng et al., 2023; Cief et al., 2024) or relaxing causal assumptions (Taufiq et al., 2024; Saito et al., 2023). A complementary line of research addresses computational challenges: training policies over large action spaces scales linearly with the number of actions $K$, motivating fast maximum inner product search (MIPS) techniques (Shrivastava & Li, 2014; Sakhi et al., 2023c) to reduce complexity.

Beyond the offline setting, several works study large action spaces in the *online* contextual bandit framework (Foster et al., 2020; Xu & Zeevi, 2020; Zhu et al., 2022; Aouali, 2025). These approaches rely on active exploration and repeated interaction with the environment, enabling algorithms to gather information adaptively. Their assumptions and techniques are therefore not applicable to the offline regime we consider, where learning must rely entirely on a fixed, biased log of historical actions. Unlike this online literature, our work investigates the optimization landscape specific to offline learning in large action spaces and provides practical and theoretical insights on how to make these offline objectives more amenable to gradient-based optimization. We view this as a fundamental yet relatively unexplored research direction.

# B. Proofs for Oracle Policies

### B.1. Oracle Policies for IPS-Based Objectives

**(IPS), cIPS and ES.** Recall the definition of the (logging propensity) clipped IPS estimator with $\tau \in [0, 1]$:

$$\hat{V}_{\text{CIPS}}(\pi) = \frac{1}{n} \sum_{i=1}^{n} \frac{\pi(A_i|X_i)}{\max\{\pi_0(A_i|X_i), \tau\}} R_i.$$

Taking $n \to \infty$, one obtains:

$$V_{\text{CIPS}}(\pi) = \mathbb{E}_{X \sim \nu, A \sim \pi_0(\cdot|X)} \left[ \frac{\pi(A|X)}{\max\{\pi_0(A|X), \tau\}} r(X, A) \right]$$

$$= \mathbb{E}_{X \sim \nu, A \sim \pi(\cdot|X)} \left[ \frac{\pi_0(A|X)}{\max\{\pi_0(A|X), \tau\}} r(X, A) \right].$$

As the objective is linear in the policy $\pi$, the optimal policy should put for any $x \in \mathcal{X}$, all the mass on the action $a$ that maximizes the weighted reward, giving:

$$\pi_*^{\texttt{CIPS}}(a|x) = \mathbb{1}\left[ a = \underset{a' \in \mathcal{A}}{\text{argmax}} \, \frac{\pi_0(a'|x) r(x, a')}{\max\{\pi_0(a'|x), \tau\}} \right].$$

We recover the solution for $\texttt{IPS}$ when we let $\tau \to 0$:

$$\pi_*^{\texttt{IPS}}(a|x) = \mathbb{1}\left[ a = \underset{a' \in \mathcal{A}}{\text{argmax}} \, r(x, a') \mathbb{1}\left[ \pi_0(a'|x) > 0 \right] \right].$$

We also recover the solution of $\texttt{ES}$ just by replacing the clipping function by an exponential function of factor $\alpha$, obtaining:

$$\pi_*^{\texttt{ES}}(a|x) = \mathbb{1}\left[ a = \underset{a' \in \mathcal{A}}{\text{argmax}} \, r(x, a') \pi_0(a'|x)^{1-\alpha} \right].$$

**Doubly Robust (DR).** The doubly robust estimator converges to the following quantity:

$$V_{\text{DR}}(\pi) = \mathbb{E}_{X \sim \nu, A \sim \pi(\cdot|X)} \left[ (r(X, A) - \hat{r}(X, A)) \frac{\pi_0(A|X)}{\max\{\pi_0(A|X), \tau\}} + \hat{r}(X, A) \right].$$

The objective is linear in $\pi$ and is thus maximized by the following deterministic decision rule:

$$\pi_*^{\texttt{DR}}(a|x) = \mathbb{1}\left[ a = \underset{a' \in \mathcal{A}}{\text{argmax}} \, \hat{r}(x, a') + (r(x, a') - \hat{r}(x, a')) \frac{\pi_0(a'|x)}{\max\{\pi_0(a'|x), \tau\}} \right]$$

**Marginalized IPS (MIPS) with clusters.** We adopt the same approach to look for the maximizer of $\texttt{MIPS}$. We generalize the clustering function $h$ to also account for context. We write down the estimator:

$$\hat{V}_{\text{MIPS}}(\pi) = \frac{1}{n} \sum_{i=1}^{n} \frac{\sum_{a'} \mathbb{1}\left[h(a', X_i) = h(A_i, X_i)\right] \pi(a'|X_i)}{\sum_{a''} \mathbb{1}\left[h(a'', X_i) = h(A_i, X_i)\right] \pi_0(a''|X_i)} R_i = \frac{1}{n} \sum_{i=1}^{n} \frac{\pi(C_i|X_i)}{\pi_0(C_i|X_i)} R_i,$$

with which, we recover when $n \to \infty$:

$$V_{\text{MIPS}}(\pi) = \mathbb{E}_{X \sim \nu, A \sim \pi_0(\cdot|X)} \left[ \frac{\sum_{a'} \mathbb{1}\left[h(a', X) = h(A, X)\right] \pi(a'|X)}{\sum_{a''} \mathbb{1}\left[h(a'', X) = h(A, X)\right] \pi_0(a''|X)} r(X, A) \right]$$

$$= \mathbb{E}_{X \sim \nu} \left[ \sum_{a} \pi_0(a|X) \frac{\sum_{a'} \mathbb{1}\left[h(a', X) = h(a, X)\right] \pi(a'|X)}{\sum_{a''} \mathbb{1}\left[h(a'', X) = h(a, X)\right] \pi_0(a''|X)} r(X, a) \right]$$

$$= \mathbb{E}_{X \sim \nu} \left[ \sum_{a'} \pi(a'|X) \sum_{a} \pi_0(a|X) \frac{\mathbb{1}\left[h(a', X) = h(a, X)\right]}{\sum_{a''} \mathbb{1}\left[h(a'', X) = h(a, X)\right] \pi_0(a''|X)} r(X, a) \right]$$

$$= \mathbb{E}_{X \sim \nu} \left[ \sum_{a'} \pi(a'|X) \mathbb{E}_{A \sim \pi_0(\cdot|X)} \left[ \frac{\mathbb{1}\left[h(a', X) = h(A, X)\right] r(X, A)}{\mathbb{E}_{A'' \sim \pi_0(\cdot|X)} \left[\mathbb{1}\left[h(A'', X) = h(A, X)\right]\right]} \right] \right].$$

The objective is linear in $\pi$, and depends on the action $a'$ through its cluster $h(a', \cdot)$ alone. This means that multiple solutions are maximizers as long as the policy chooses the best cluster $c$. We thus write down the oracle policy for $\texttt{MIPS}$ in the cluster level, giving:

$$
\begin{aligned}
\pi_*^{\texttt{MIPS}}(c|x) &= \mathbb{1}\Big[c = \operatorname*{argmax}_{c' \in \mathcal{C}} \Big\{ \mathbb{E}_{A \sim \pi_0(\cdot|x)} \Big[ \frac{r(x, A)\mathbb{1}[h(A, x) = c']}{\mathbb{E}_{A'' \sim \pi_0(\cdot|x)}[\mathbb{1}[h(A'', x) = h(A, x)]]} \Big] \Big\} \Big] \\
&= \mathbb{1}\Big[c = \operatorname*{argmax}_{c' \in \mathcal{C}} \Big\{ \mathbb{E}_{A \sim \pi_0(\cdot|x)} \Big[ \frac{r(x, A)\mathbb{1}[h(A, x) = c']}{\mathbb{E}_{A'' \sim \pi_0(\cdot|x)}[\mathbb{1}[h(A'', x) = c']]} \Big] \Big\} \Big] \\
&= \mathbb{1}\Big[c = \operatorname*{argmax}_{c' \in \mathcal{C}} \Big\{ \frac{\mathbb{E}_{A \sim \pi_0(\cdot|x)}[r(x, A)\mathbb{1}[h(A, x) = c']]}{\mathbb{E}_{A \sim \pi_0(\cdot|x)}[\mathbb{1}[h(A, x) = c']]} \Big\} \Big],
\end{aligned}
$$

which ends the proof.

**Conjunct Effect Modeling (OffCEM).** This estimator can be seen as the natural, doubly robust extension of the $\texttt{MIPS}$ estimator. Combining similar techniques to the ones employed for $\texttt{MIPS}$ and $\texttt{DR}$ yields

$$
\pi_*^{\texttt{OffCEM}}(a|x) = \mathbb{1}\Big[a = \operatorname*{argmax}_{a' \in \mathcal{A}} \Big\{ \hat{r}(x, a') + \frac{\mathbb{E}_{\bar{A} \sim \pi_0(\cdot|x)}[(r(x, \bar{A}) - \hat{r}(x, \bar{A}))\mathbb{1}[h(x, \bar{A}) = h(x, a')]]}{\pi_0(h(x, a')|x)} \Big\} \Big].
$$

**Two Stage Decomposition (POTEC).** This is an *optimization strategy* for $\texttt{OffCEM}$. It restricts the policy to a cluster-informed form,

$$
\pi(a \mid x) = \sum_{c \in \mathcal{C}} \pi^{\texttt{RM}}(a \mid x, c)\pi^{\texttt{CL}}(c \mid x),
$$

where $\pi^{\texttt{RM}}(a \mid x, c) = \mathbb{1}[a = \operatorname*{argmax}_{a' \in c} \hat{r}(x, a')]$ is fixed, model-based policy that deterministically selects the best action within each cluster. Learning is then simplified to finding the optimal cluster-level policy $\pi^{\texttt{CL}}$ that maximizes the $\texttt{OffCEM}$ objective:

$$
\hat{V}_{\texttt{POTEC}}(\pi^{\texttt{CL}}) = \frac{1}{n}\sum_{i=1}^{n}\Big( \frac{\pi^{\texttt{CL}}(C_i \mid X_i)}{\pi_0(C_i \mid X_i)}(R_i - \hat{r}(X_i, A_i)) + \sum_{c \in \mathcal{C}}\pi^{\texttt{CL}}(c \mid X_i)\hat{r}_c^*(X_i) \Big),
$$

where $\hat{r}_c^*(x) = \max_{a \in c}\hat{r}(x, a)$ is the estimated reward of the best action in cluster $c$. This is exactly the Doubly Robust version of $\texttt{MIPS}$ on the cluster level, the oracle policy on the cluster level can be followed in the same fashion:

$$
\pi_*^{\texttt{CL}}(c \mid x) = \mathbb{1}\Big[c = \operatorname*{argmax}_{c' \in \mathcal{C}} \Big\{ \frac{\mathbb{E}_{A \sim \pi_0(\cdot|x)}[(r(x, A) - \hat{r}(x, A))\mathbb{1}[h(A, x) = c']]}{\mathbb{E}_{A \sim \pi_0(\cdot|x)}[\mathbb{1}[h(A, x) = c']]} + \hat{r}_{c'}^*(x) \Big\} \Big].
$$

The optimal policy for the $\texttt{POTEC}$ optimization strategy unfolds as:

$$
\pi_*^{\texttt{POTEC}}(a|x) = \sum_{c \in \mathcal{C}} \pi^{\texttt{RM}}(a \mid x, c)\pi_*^{\texttt{CL}}(c \mid x).
$$

At first glance, it might be hard to see the connection between $\texttt{POTEC}$ and $\texttt{OffCEM}$ solutions, but they are equivalent. For ease of notation, let us denote by $D_{\hat{r},x}(c)$:

$$
D_{\hat{r},x}(c) = \frac{\mathbb{E}_{A \sim \pi_0(\cdot|x)}[(r(x, A) - \hat{r}(x, A))\mathbb{1}[h(A, x) = c]]}{\mathbb{E}_{A \sim \pi_0(\cdot|x)}[\mathbb{1}[h(A, x) = c]]}.
$$

and recall that the optimal policy of $\texttt{OffCEM}$ finds the action $a$ that maximizes:

$$
\tilde{V}(x, a) = \hat{r}(x, a) + D_{\hat{r},x}(h(a, x)).
$$

For any context $x$, the optimal action $a^*$ of $\texttt{POTEC}$ verifies:

- $a^*$ is in the optimal cluster: $h(a^*, x) = c_*(x)$ with $c_*(x) = \operatorname{argmax}_{c \in \mathcal{C}} D_{\hat{r},x}(c) + \hat{r}_c^*(x)$.

- $a^*$ is optimal within that cluster: $a = \mathrm{argmax}_{a \in c_*(x)} \hat{r}(x, a)$.

This means that for all actions $a$ with $h(a, x) \neq c_*(x)$, we have:

$$
\begin{aligned}
\tilde{V}(x, a) &= D_{\hat{r},x}(h(a, x)) + \hat{r}(x, a) \\
&\leq D_{\hat{r},x}(h(a, x)) + \hat{r}^*_{h(a,x)}(x) \\
&\leq D_{\hat{r},x}(c_*(x)) + \hat{r}^*_{c_*(x)}(x) \\
&= D_{\hat{r},x}(h(x, a^*)) + \hat{r}(x, a^*) = \tilde{V}(x, a^*).
\end{aligned}
$$

In addition, for all actions $a$ with $h(a, x) = c_*(x)$, we have:

$$
\begin{aligned}
\tilde{V}(x, a) &= D_{\hat{r},x}(h(a, x)) + \hat{r}(x, a) \\
&= D_{\hat{r},x}(c_*(x)) + \hat{r}(x, a) \\
&\leq D_{\hat{r},x}(c_*(x)) + \hat{r}^*_{c_*(x)}(x) = \tilde{V}(x, a^*).
\end{aligned}
$$

This means that the optimal action $a^*$ for `POTEC` is the maximizer of $\tilde{V}(x, a)$, which is exactly the solution of `OffCEM`.

**Policy Convolution (PC).** This estimator uses a nearest neighbors function to aggregate the propensities of similar actions, making the hypothesis that similar actions will result in similar reward signal. The estimator writes:

$$
\hat{V}_{\mathrm{PC}}(\pi) = \frac{1}{n} \sum_{i=1}^{n} \frac{\pi(N_\epsilon(A_i) \mid X_i)}{\pi_0(N_\epsilon(A_i) \mid X_i)} R_i, \quad \text{with } \pi(N_\epsilon(a) \mid x) = \sum_{a' \in N_\epsilon(a)} \pi(a' \mid x).
$$

This estimator is equivalent to the following when $n \to \infty$:

$$
\begin{aligned}
V^{\mathrm{PC}}(\pi) &= \mathbb{E}_{X \sim \nu, A \sim \pi_0(\cdot|X)} \left[ \frac{\sum_{a'} \pi(a'|X) \mathbb{1}\left[a' \in N_\epsilon(A)\right]}{\pi_0(N_\epsilon(A)|X)} r(X, A) \right] \\
&= \mathbb{E}_{X \sim \nu, A \sim \pi(\cdot|X)} \left[ \mathbb{E}_{\bar{A} \sim \pi_0(\cdot|X)} \left[ \frac{r(x, \bar{A}) \mathbb{1}\left[A \in N_\epsilon(\bar{A})\right]}{\pi_0(N_\epsilon(\bar{A})|X)} \right] \right].
\end{aligned}
$$

The same argument of linearity applies here, giving us the corresponding oracle policy:

$$
\pi^{\mathrm{PC}}_*(a|x) = \mathbb{1}\left[ a = \mathrm{argmax}_{a' \in \mathcal{A}} \left\{ \mathbb{E}_{\bar{A} \sim \pi_0(\cdot|x)} \left[ \frac{r(x, \bar{A}) \mathbb{1}[a' \in N_\epsilon(\bar{A})]}{\pi_0(N_\epsilon(\bar{A})|x)} \right] \right\} \right].
$$

### B.2. Oracle Policies for PWLL-Based Objectives

Our objectives can be written in the same form, only choosing for each a different function $g$:

$$
\hat{U}_g(\pi) = \frac{1}{n} \sum_{i=1}^{n} g(X_i, A_i, R_i) \log \pi(A_i \mid X_i).
$$

Since we are looking at oracle policies, we consider the expectation

$$
U_g(\pi) = \mathbb{E}_{X \sim \nu, A \sim \pi_0(\cdot|X), R \sim p(\cdot|X,A)} \left[ g(X, A, R) \log \pi(A \mid X) \right].
$$

The maximization decomposes over contexts. Fix $x$ and define the nonnegative weights

$$
w_x(a) = \mathbb{E}_{R \sim p(\cdot|x,a)} \left[ g(x, a, R) \right] \geq 0.
$$

For each $x$, we thus consider

$$
\max_{\pi(\cdot|x)} \sum_{a \in \mathcal{A}} \pi_0(a \mid x) \, w_x(a) \, \log \pi(a \mid x)
$$

$$
\text{s.t.} \quad \sum_{a \in \mathcal{A}} \pi(a \mid x) = 1, \qquad \forall a \in \mathcal{A}, \, \pi(a \mid x) \geq 0.
$$

Let $v_x(a) = \pi_0(a \mid x)\, w_x(a) \geq 0$. The Lagrangian (with equality multiplier $\lambda \in \mathbb{R}$ and inequality multipliers $\{\mu(a)\}_{a \in \mathcal{A}}$, $\mu(a) \geq 0$) is

$$\mathcal{L}(\pi, \lambda, \mu) = \sum_{a \in \mathcal{A}} v_x(a) \log \pi(a \mid x) + \lambda \Big( \sum_{a \in \mathcal{A}} \pi(a \mid x) - 1 \Big) + \sum_{a \in \mathcal{A}} \mu(a)\, \pi(a \mid x).$$

By KKT conditions, at an optimum $\pi_*^g(\cdot \mid x)$ we have for all $a \in \mathcal{A}$:

$$\frac{\partial \mathcal{L}}{\partial \pi(a \mid x)} = \frac{v_x(a)}{\pi(a \mid x)} + \lambda + \mu(a) = 0, \qquad \text{and} \quad \mu(a)\, \pi(a \mid x) = 0.$$

For any action with $\pi_*^g(a \mid x) > 0$, we get that $\mu(a) = 0$, and hence

$$\pi_*^g(a \mid x) = -\frac{v_x(a)}{\lambda}.$$

Normalizing with $\sum_a \pi_*^g(a \mid x) = 1$ gives $\lambda = -\sum_{a'} v_x(a')$ and therefore

$$\pi_*^g(a \mid x) = \frac{v_x(a)}{\sum_{a' \in \mathcal{A}} v_x(a')} = \frac{\pi_0(a \mid x)\, \mathbb{E}_{R \sim p(\cdot \mid x,a)}\,[g(x, a, R)]}{\sum_{a' \in \mathcal{A}} \pi_0(a' \mid x)\, \mathbb{E}_{R \sim p(\cdot \mid x,a')}\,[g(x, a', R)]}.$$

This concludes the proof.

## C. Proofs for Optimization Properties

In this section, we prove the propositions about the optimization landscape of IPS-based and PWLL learning approaches. We start by stating the following lemmas, that will be helpful to prove our propositions.

**Lemma C.1.** *([Mei et al., 2020b](), Lemma 2) Consider the single context case. With a slight abuse of notation, we drop the dependence on $x$ and write $r(a)$ instead of $r(x, a)$, $\pi_\theta(a)$ instead of $\pi_\theta(a \mid x)$, and $\hat{r}(a)$ instead of $\hat{r}(x, a)$. Let $\pi_\theta$ be a softmax policy parameterized by $\theta$. Then, for any $\hat{r} \in [0, 1]^K$, and any estimator $\hat{V}$ linear in $\pi_\theta$, the mapping $\theta \mapsto \hat{V}(\pi_\theta) = \langle \hat{r}, \pi_\theta \rangle$ is $5/2$-smooth.*

**Lemma C.2.** *All the action level estimators* EST *in (* IPS*,* cIPS*,* DR*,* PC*) can be written, for any policy $\pi$, in the form:*

$$\hat{V}_{\text{EST}}(\pi) = \frac{1}{n} \sum_{i=1}^n \mathbb{E}_{A \sim \pi(\cdot \mid X_i)}\,[\hat{r}_{\text{EST},i}(A, X_i)]\,, \tag{16}$$

*For the cluster level estimators/approaches* EST-C *in (* MIPS*,* OffCEM*,* POTEC*), we also have*

$$\hat{V}_{\text{EST-C}}(\pi) = \frac{1}{n} \sum_{i=1}^n \mathbb{E}_{C \sim \pi(\cdot \mid X_i)}\,[\hat{r}_{\text{EST-C},i}(C, X_i)]\,, \tag{17}$$

*meaning that all these estimators are linear in $\pi$.*

*Proof.* This is straightforward to prove. We begin by the action level estimators and take DR as a representative. For DR, we have the following:

$$\hat{r}_{\text{DR},i}(a, X_i) = \hat{r}(a, X_i) + \mathbb{I}[a = A_i] \frac{R_i - \hat{r}(A_i, X_i)}{\max(\tau, \pi_0(A_i \mid X_i))}$$

verifies the equation. Solutions for cIPS and IPS can be recovered directly, and PC follows the same construction. For the cluster level approaches, we take POTEC as a representative, and we have:

$$\hat{r}_{\text{POTEC},i}(c, X_i) = \hat{r}_c^\star(X_i) + \mathbb{I}[c = C_i] \frac{R_i - \hat{r}(A_i, X_i)}{\pi_0(C_i \mid X_i)}\,,$$

The $\hat{r}_{\text{MIPS},i}$ follows as a special case when $\hat{r} = 0$. $\qquad \square$

**Lemma C.3.** *Consider the single-context case and assume a finite action set $\mathcal{A}$. For any estimator EST in (IPS, cIPS, DR, OffCEM, MIPS, PC), there exists a problem instance (i.e., a choice of $r$ and $\pi_0$; and when relevant, a choice of auxiliary objects such as $\hat{r}$, $h$, $N_\epsilon$) such that, in the large-$n$ limit,*

$$\hat{r}_{\text{EST}}(a) = \mathbb{1}[a = a_K]$$

*for some optimal action $a_K$. Similarly, for cluster-based approaches (e.g., POTEC and MIPS), there exists an instance such that*

$$\hat{r}_{\text{EST-C}}(c) = \mathbb{1}[c = c_{|\mathcal{C}|}]$$

*for some optimal cluster $c_{|\mathcal{C}|}$.*

*Proof.* We give explicit constructions for cIPS (action-level) and POTEC (cluster-level). The other estimators follow by the same idea: choose a setting where the estimator becomes linear in $\pi$ with some deterministic coefficient, and pick $r$ (and possibly $\hat{r}$, $h$, $N_\epsilon$) so that the resulting linearized reward is one-hot.

**Action-level: cIPS.** Fix $\tau \in [0,1)$.[4] Choose a logging policy $\pi_0$ with full support and such that

$$\max_{a \in \mathcal{A}} \pi_0(a) \geq \tau, \tag{18}$$

Let

$$a_K \in \arg\max_{a \in \mathcal{A}} \frac{\pi_0(a)}{\max\{\pi_0(a), \tau\}}.$$

Under Equation (18), there exists at least one action with $\pi_0(a) \geq \tau$, for which the ratio equals 1, hence the maximizer satisfies $\pi_0(a_K) \geq \tau$ and therefore

$$\frac{\pi_0(a_K)}{\max\{\pi_0(a_K), \tau\}} = 1.$$

Now define the reward function

$$r(a) = \mathbb{1}[a = a_K] \frac{\max\{\pi_0(a), \tau\}}{\pi_0(a)}.$$

This satisfies $r(a) \in [0,1]$ for all $a$ because $r(a) = 0$ for $a \neq a_K$, and

$$r(a_K) = \frac{\max\{\pi_0(a_K), \tau\}}{\pi_0(a_K)} = 1 \quad (\text{since } \pi_0(a_K) \geq \tau).$$

For cIPS, the large-$n$ linearized reward is

$$\hat{r}_{\text{cIPS}}(a) = \frac{\pi_0(a)}{\max\{\pi_0(a), \tau\}} r(a),$$

hence

$$\hat{r}_{\text{cIPS}}(a) = \frac{\pi_0(a)}{\max\{\pi_0(a), \tau\}} \mathbb{1}[a = a_K] \frac{\max\{\pi_0(a), \tau\}}{\pi_0(a)} = \mathbb{1}[a = a_K],$$

as desired.

**Cluster-level: POTEC.** We work in the single-context case and consider a clustering map $h : \mathcal{A} \to \mathcal{C}$. Choose $h$ so that $a_K$ forms a singleton cluster:

$$c_{|\mathcal{C}|} = h(a_K) = \{a_K\}, \qquad h(a) \neq c_{|\mathcal{C}|} \;\; \forall a \neq a_K.$$

Let rewards be $r(a_K) = 1$ and $r(a) = 0$ for $a \neq a_K$. Pick any $\varepsilon \in (0, 1/2]$ and define a reward model

$$\hat{r}(a_K) = 1 - \varepsilon, \qquad \hat{r}(a) = \varepsilon \;\; \forall a \neq a_K.$$

_______________

[4]If $\tau = 1$ and $|\mathcal{A}| > 1$, the simplifying assumption $\pi_0(a) > 0$ for all $a$ is incompatible with having some $\pi_0(a) \geq \tau$. In practice $\tau \ll 1$.

For `POTEC`, the induced (cluster-level) linearized reward takes the form

$$\hat{r}_{\text{POTEC}}(c) = \max_{a \in c} \hat{r}(a) + \frac{\sum_{a \in c} \pi_0(a)\big(r(a) - \hat{r}(a)\big)}{\pi_0(c)}.$$

For the singleton cluster $c_{|\mathcal{C}|} = \{a_K\}$, we get

$$\hat{r}_{\text{POTEC}}(c_{|\mathcal{C}|}) = (1 - \varepsilon) + \frac{\pi_0(a_K)\big(1 - (1 - \varepsilon)\big)}{\pi_0(a_K)} = (1 - \varepsilon) + \varepsilon = 1.$$

For any other cluster $c \neq c_{|\mathcal{C}|}$, all its actions satisfy $r(a) = 0$ and $\hat{r}(a) = \varepsilon$, hence

$$\hat{r}_{\text{POTEC}}(c) = \varepsilon + \frac{\sum_{a \in c} \pi_0(a)(-\varepsilon)}{\pi_0(c)} = \varepsilon - \varepsilon = 0.$$

Therefore $\hat{r}_{\text{POTEC}}(c) = \mathbb{1}[c = c_{|\mathcal{C}|}]$.

This concludes the constructions for `cIPS` and `POTEC`. The remaining estimators can be handled analogously by choosing $\pi_0$ (and when relevant, $h$ or $N_e$) so that the estimator's linear coefficient on $r(a)$ equals 1 at a chosen $a_K$ (and equals something finite elsewhere), and then defining $r$ (and possibly $\hat{r}$) to make the resulting $\hat{r}_{\text{EST}}$ one-hot. □

Now we restate Proposition 2.1 and proceed to its proof.

**Proposition C.4** (Plateau for linear-in-$\pi$ objectives under softmax). *Consider the single-context case. Let $\hat{V}(\pi)$ be any objective linear in $\pi$ and let $\pi^\star \in \arg\max_\pi \hat{V}(\pi)$ denote a maximizer over the probability simplex on the effective action space $\mathcal{A}_{\text{eff}}$ (of size $K_{\text{eff}} = |\mathcal{A}_{\text{eff}}|$). Let $\{\pi_{\theta_t}\}_{t \geq 1}$ be the iterates of gradient ascent on $\theta \mapsto \hat{V}(\pi_\theta)$ with a linear softmax policy $\pi_\theta(a) = \exp(\theta_a)/\sum_{a' \in \mathcal{A}_{\text{eff}}} \exp(\theta_{a'})$ and step sizes $\eta_t \in (0, 1]$. Then there exists a problem instance such that gradient ascent cannot escape a suboptimal region before $t_0 = C K_{\text{eff}} = \mathcal{O}(K_{\text{eff}})$ iterations, in the sense that*

$$\forall t \leq t_0 : \qquad \hat{V}(\pi^\star) - \hat{V}(\pi_{\theta_t}) \geq 0.9.$$

*Proof.* The proof follows the same technique as (Mei et al., 2020a, Theorem 1). By Lemma C.3, there exists an instance (single context) for which the linearized reward is one-hot:

$$\hat{r}_{\text{EST}}(a) = \mathbb{1}[a = a_K] \qquad \text{for some } a_K \in \mathcal{A}_{\text{eff}}.$$

Hence, for any policy $\pi$ supported on $\mathcal{A}_{\text{eff}}$,

$$\hat{V}(\pi) = \sum_{a \in \mathcal{A}_{\text{eff}}} \pi(a)\hat{r}_{\text{EST}}(a) = \pi(a_K).$$

The maximizer over the simplex is therefore $\pi^\star = \delta_{a_K}$, and

$$\hat{V}(\pi^\star) = 1, \qquad \hat{V}(\pi^\star) - \hat{V}(\pi_\theta) = 1 - \pi_\theta(a_K).$$

(Notice that $\sup_\theta \hat{V}(\pi_\theta) = 1$ as well, although the supremum is not attained by any finite $\theta$ when $K_{\text{eff}} \geq 2$.) We now upper bound the gradient norm. For the softmax parametrization,

$$\left\|\nabla_\theta \hat{V}(\pi_\theta)\right\|_2 \leq \sqrt{2}\,\pi_\theta(a_K)\big(1 - \pi_\theta(a_K)\big),$$

where the bound follows by a direct computation (as in (Mei et al., 2020a)).

Define the update $\theta_{t+1} = \theta_t + \eta_t \nabla_\theta \hat{V}(\pi_{\theta_t})$ and split iterations into

$$t_{\text{good}} = \{t \geq 1 : \pi_{\theta_{t+1}}(a_K) > \pi_{\theta_t}(a_K)\}, \qquad t_{\text{bad}} = \{t \geq 1 : \pi_{\theta_{t+1}}(a_K) \leq \pi_{\theta_t}(a_K)\}.$$

For $t \in t_{\text{bad}}$,

$$\frac{1}{\pi_{\theta_t}(a_K)} - \frac{1}{\pi_{\theta_{t+1}}(a_K)} \leq 0.$$

For $t \in t_{\text{good}}$, using Lemma C.1 (the $5/2$-smoothness of $\theta \mapsto \hat{V}(\pi_\theta)$) and $\eta_t \in (0,1]$, we obtain

$$\pi_{\theta_{t+1}}(a_K) - \pi_{\theta_t}(a_K) \leq \frac{9}{2}\pi_{\theta_t}(a_K)^2,$$

and therefore (since $\pi_{\theta_{t+1}}(a_K) \geq \pi_{\theta_t}(a_K) > 0$),

$$\frac{1}{\pi_{\theta_t}(a_K)} - \frac{1}{\pi_{\theta_{t+1}}(a_K)} = \frac{\pi_{\theta_{t+1}}(a_K) - \pi_{\theta_t}(a_K)}{\pi_{\theta_{t+1}}(a_K)\pi_{\theta_t}(a_K)} \leq \frac{9}{2}.$$

Summing over $s = 1, \ldots, t-1$ yields

$$\frac{1}{\pi_{\theta_1}(a_K)} - \frac{1}{\pi_{\theta_t}(a_K)} = \sum_{s=1}^{t-1}\left(\frac{1}{\pi_{\theta_s}(a_K)} - \frac{1}{\pi_{\theta_{s+1}}(a_K)}\right) \leq \frac{9}{2}t.$$

Assume a standard symmetric initialization so that $\pi_{\theta_1}(a_K) = 1/K_{\text{eff}}$. Pick any constant $c \geq 11$ and take $K_{\text{eff}}$ large enough so that $\pi_{\theta_1}(a_K) \leq 1/c$. If $t \leq \frac{2}{9c}K_{\text{eff}}$, then

$$\frac{1}{\pi_{\theta_t}(a_K)} \geq \frac{1}{\pi_{\theta_1}(a_K)} - \frac{9}{2}t \geq \frac{1}{\pi_{\theta_1}(a_K)}\left(1 - \frac{1}{c}\right) \geq c - 1 \geq 10,$$

hence $\pi_{\theta_t}(a_K) \leq 1/10$, and thus

$$\hat{V}(\pi^\star) - \hat{V}(\pi_{\theta_t}) = 1 - \pi_{\theta_t}(a_K) \geq 0.9.$$

This proves the claim with $t_0 = \frac{2}{9c}K_{\text{eff}}$. $\qquad\square$

**Proposition C.5.** *Even for a single context $x$, deterministic rewards, there is problem where IPS-based learning with a linear softmax policy $\pi_\theta(a) \propto \exp(\langle\theta, \phi(x,a)\rangle)\mathbb{I}[a \in \mathcal{A}_{\textit{eff}}]$ can have a number of local maxima exponential in the number of effective actions $K_{\textit{eff}}$.*

*Proof.* Let EST an off-policy estimators considered in the paper with an action-level policy. By Lemma C.2, we have:

$$\hat{V}_{\text{EST}}(\pi) = \frac{1}{n}\sum_{i=1}^{n}\mathbb{E}_{a \sim \pi(\cdot|x_i)}\left[\hat{r}_{\text{EST},i}(a, x_i)\right], \tag{19}$$

In a single context setting, it becomes:

$$\hat{V}_{\text{EST}}(\pi_\theta) = \mathbb{E}_{a \sim \pi_\theta(\cdot)}\left[\frac{1}{n}\sum_{i=1}^{n}\hat{r}_{\text{EST},i}(a)\right], \tag{20}$$

$$= \langle\frac{1}{n}\sum_{i=1}^{n}\hat{r}_{\text{EST},i}, \pi_\theta\rangle. \tag{21}$$

This also holds for estimators with policies in the cluster level, as we still have:

$$\hat{V}_{\text{EST-C}}(\pi_\theta) = \mathbb{E}_{c \sim \pi_\theta(\cdot)}\left[\frac{1}{n}\sum_{i=1}^{n}\hat{r}_{\text{EST-C},i}(c)\right], \tag{22}$$

$$= \langle\frac{1}{n}\sum_{i=1}^{n}\hat{r}_{\text{EST-C},i}, \pi_\theta\rangle. \tag{23}$$

These softmax policies are all defined on the effective action space $\mathcal{A}_{\text{eff}}$, be it a subset of the action space $\mathcal{A}$ or the discrete cluster space $\mathcal{C}$. Using the linearity of the objective, we can directly apply Theorem 1 from (Chen et al., 2019) and obtain our result. $\qquad\square$

Finally, we also restate Proposition 3.1, and provide its proof.

**Proposition C.6.** *For an $\ell_2$ regularized (substituting $\frac{\lambda}{2}||\theta||^2$, with $\lambda > 0$), linear softmax policy $\pi_\theta$, the PWLL objective $\hat{U}^g(\pi_\theta)$ defined as:*

$$\hat{U}^g(\pi) = \frac{1}{n}\sum_{i=1}^{n} g(R_i, \pi_0(A_i \mid X_i)) \log \pi(A_i \mid X_i),$$

*is $\lambda$-strongly concave. Without regularization, the objective is concave.*

*Proof.* For any $x$ and $a \in \mathcal{A}_{\text{eff}}(x)$, we have:

$$\pi_\theta(a|x) = \frac{\exp(\langle \theta, \phi(x, a)\rangle)}{\sum_{a' \in \mathcal{A}_{\text{eff}}(x)} \exp(\langle \theta, \phi(x, a')\rangle)},$$

optimizing an $\ell_2$ regularized linear softmax, giving:

$$\hat{L}^{g,\lambda}(\pi) = \hat{U}^g(\pi) - \frac{\lambda}{2}||\theta||^2,$$

with $\lambda > 0$ and recall that $g \geq 0$. For strong concavity, we need to show that the Hessian $\nabla^2_\theta \hat{U}^g(\pi_\theta)$ is negative definite with eigenvalues bounded away from zero.

The gradient with respect to $\theta$ is: $\nabla_\theta \hat{U}^g(\pi_\theta) = \frac{1}{n}\sum_{i=1}^{n} g(R_i, \pi_0(A_i \mid X_i))\nabla_\theta \log \pi_\theta(A_i|X_i) - \lambda\theta$

For the softmax policy:

$$\nabla_\theta \log \pi_\theta(a|x) = \phi(x, a) - \sum_{a'} \pi_\theta(a'|x)\phi(x, a') = \phi(x, a) - \mathbb{E}_{A\sim\pi_\theta(\cdot|x)}[\phi(x, A)]$$

Therefore: $\nabla_\theta \hat{U}^g(\pi_\theta) = \frac{1}{n}\sum_{i=1}^{n} g(R_i, \pi_0(A_i \mid X_i))\left(\phi(X_i, A_i) - \mathbb{E}_{A\sim\pi_\theta(\cdot|X_i)}[\phi(X_i, A)]\right) - \lambda\theta$

Taking the second derivative: $\nabla^2_\theta \hat{U}^g(\pi_\theta) = -\frac{1}{n}\sum_{i=1}^{n} g(R_i, \pi_0(A_i \mid X_i))\nabla_\theta \mathbb{E}_{A\sim\pi_\theta(\cdot|X_i)}[\phi(X_i, A)] - \lambda I_d$, where $I_d$ is the $d \times d$ identity matrix. The gradient of the expectation is:

$$\nabla_\theta \mathbb{E}_{A\sim\pi_\theta(\cdot|x)}[\phi(x, A)] = \sum_a \nabla_\theta \pi_\theta(a|x)\phi(x, a)$$

Using $\nabla_\theta \pi_\theta(a|x) = \pi_\theta(a|x)(\phi(x, a) - \mathbb{E}_{A\sim\pi_\theta(\cdot|x)}[\phi(x, A)])$:

$$\nabla_\theta \mathbb{E}_{A\sim\pi_\theta(\cdot|x)}[\phi(x, A)] = \sum_a \pi_\theta(a|x)(\phi(x, a) - \mathbb{E}_{A\sim\pi_\theta(\cdot|x)}[\phi(x, A)])\phi(x, a)^\top$$

This simplifies to:

$$\nabla_\theta \mathbb{E}_{A\sim\pi_\theta(\cdot|x)}[\phi(x, A)] = \text{Cov}_{A\sim\pi_\theta(\cdot|x)}[\phi(x, A)]$$

where $\text{Cov}_{A\sim\pi_\theta(\cdot|x)}[\phi(x, A)] = \mathbb{E}_{A\sim\pi_\theta(\cdot|x)}[\phi(x, A)\phi(x, A)^\top] - \mathbb{E}_{A\sim\pi_\theta(\cdot|x)}[\phi(x, A)]\mathbb{E}_{A\sim\pi_\theta(\cdot|x)}[\phi(x, A)]^\top$

Therefore:

$$\nabla^2_\theta \hat{U}^g(\pi_\theta) = -\frac{1}{n}\sum_{i=1}^{n} g(R_i, \pi_0(A_i \mid X_i))\text{Cov}_{A\sim\pi_\theta(\cdot|X_i)}[\phi(X_i, A)] - \lambda I_d$$

We can write this as: $\nabla^2_\theta \hat{U}^g(\pi_\theta) = -H - \lambda I_d$

where $H = \frac{1}{n}\sum_{i=1}^{n} g(R_i, \pi_0(A_i \mid X_i))\text{Cov}_{A\sim\pi_\theta(\cdot|X_i)}[\phi(X_i, A)]$ is positive semi-definite. To see this explicitly, for any vector $v \in \mathbb{R}^d$:

$$v^\top \text{Cov}_{A\sim\pi_\theta(\cdot|X_i)}[\phi(X_i, A)]v = \text{Var}_{A\sim\pi_\theta(\cdot|X_i)}[v^\top \phi(X_i, A)] \geq 0,$$

with the positivity of $g$, this ensures $H$ is positive semi-definite. Then we have:

$$v^\top \nabla_\theta^2 \hat{U}^g(\pi_\theta)v = -v^\top Hv - \lambda v^\top v = -v^\top Hv - \lambda\|v\|^2\,,$$

meaning that when $v \neq 0$, we get $v^\top \nabla_\theta^2 \hat{U}^g(\pi_\theta)v \leq -\lambda\|v\|^2 < 0$.

This shows the Hessian is negative definite with all eigenvalues bounded above by $-\lambda < 0$. Therefore, $\ell_2$ regularized $\hat{U}^g(\pi_\theta)$ is $\lambda$-strongly concave. In addition, when $\lambda = 0$, the hessian is negative semi-definite, giving simple concavity. $\qquad\square$

## D. Stochastic Optimization Convergence Guarantees for PWLL

We analyze the convergence rates of stochastic gradient methods on the PWLL objective. We formulate this as the minimization of the finite-sum loss $f(\theta) = -\hat{U}_g(\pi_\theta)$:

$$f(\theta) = \frac{1}{n}\sum_{i=1}^{n} f_i(\theta), \quad \text{where } f_i(\theta) = -g_i \log \pi_\theta(A_i \mid X_i), \tag{24}$$

where $g_i = g(R_i, \pi_0(A_i|X_i))$. We adopt the linear softmax policy parametrization in Equation (10) with $s_\theta(x,a) = \phi(x,a)^\top\theta$ (lightweight parametrization in Equation (11)). We note that our analysis extends naturally to the heavyweight parametrization in Equation (11).

### D.1. Assumptions and Regularity

To establish problem-dependent convergence bounds, we rely on the following structural assumptions regarding the feature space and the importance weights.

**Assumption D.1** (Bounded features). For all context-action pairs $(x,a) \in \mathcal{X} \times \mathcal{A}$, the feature representations are bounded in Euclidean norm:

$$\|\phi(x,a)\|_2 \leq H.$$

**Assumption D.2** (Bounded weighting function). The weights $g_i = g(R_i, \pi_0(A_i|X_i))$ computed on the static dataset are strictly positive and bounded. That is, for all $i \in \{1, \ldots, n\}$:

$$0 < g_i \leq G_{\max}.$$

Assumptions D.1 and D.2 are sufficient to establish the smoothness and bounded variance of the objective $f(\theta)$. We formally derive these properties in the following proposition.

**Proposition D.3** (Regularity and Variance Bounds). *Under Assumptions D.1 and D.2, the objective $f(\theta)$ satisfies the following properties:*

1. **Global Smoothness:** *The objective is $\bar{L}$-smooth with $\bar{L} = G_{\max}H^2$.*

2. **Bounded Single-Sample Variance:** *The variance of the stochastic gradient for a single sample is bounded by $\bar{\sigma}^2 = 4G_{\max}^2 H^2$.*

3. **Bounded Mini-Batch Variance:** *For a mini-batch of size $b$, the variance is bounded by $\bar{\sigma}_b^2 = \frac{4G_{\max}^2 H^2}{b}$.*

*Proof. 1. Smoothness:* The Hessian of the objective is the weighted sum of the feature covariance matrices under the policy $\pi_\theta$:

$$\nabla^2 f(\theta) = \frac{1}{n}\sum_{i=1}^{n} g_i \text{Cov}_{A\sim\pi_\theta(\cdot|X_i)}[\phi(X_i, A)].$$

The spectral norm of a covariance matrix is bounded by the maximum squared norm of its random vectors. Thus, using Theorem D.1 we get that $\|\nabla^2 f(\theta)\|_{\text{op}} \leq \frac{1}{n}\sum_{i=1}^{n} g_i H^2 \leq G_{\max}H^2$.

*2. Single-Sample Variance:* We first bound the norm of the gradient for an arbitrary sample $i$. The gradient is $\nabla f_i(\theta) = -g_i(\phi(X_i, A_i) - \mathbb{E}_{A\sim\pi_\theta(\cdot|X_i)}[\phi(X_i, A)])$. Using the triangle inequality and Assumption D.1:

$$\|\nabla f_i(\theta)\|_2 \leq g_i \left(\|\phi(X_i, A_i)\|_2 + \|\mathbb{E}_{A\sim\pi_\theta(\cdot|X_i)}[\phi(X_i, A)]\|_2\right) \leq G_{\max}(H + H) = 2G_{\max}H.$$

Let $\xi = \nabla f_I(\theta)$ be the stochastic gradient sampled uniformly from the dataset. The variance is bounded by the second moment:

$$\mathrm{Var}(\xi) \leq \mathbb{E}[\|\xi\|^2] = \frac{1}{n}\sum_{i=1}^{n}\|\nabla f_i(\theta)\|^2 \leq (2G_{\max}H)^2 = 4G_{\max}^2 H^2.$$

*3. Mini-Batch Variance:* Let the mini-batch gradient be $\bar{g}_t = \frac{1}{b}\sum_{j=1}^{b}\nabla f_{i_j}(\theta)$, where indices are sampled independently with replacement. Using the standard variance reduction property for independent variables:

$$\mathbb{E}[\|\bar{g}_t - \nabla f(\theta)\|^2] = \frac{1}{b}\mathbb{E}[\|\nabla f_I(\theta) - \nabla f(\theta)\|^2] \leq \frac{4G_{\max}^2 H^2}{b}.$$

$\square$

Based on Proposition D.3, we define the following global problem-dependent constants on which our convergence rates depend:

- $\bar{L} = G_{\max}H^2$: Smoothness constant.

- $\bar{\sigma}^2 = 4G_{\max}^2 H^2$: Upper bound on the gradient variance for a single sample.

- $\bar{\sigma}_b^2 = \frac{4G_{\max}^2 H^2}{b}$: Upper bound on the gradient variance for a mini-batch of size $b$.

## D.2. PWLL without $\ell_2$ regularization

We begin by analyzing the standard unregularized PWLL objective. Here, the objective $f(\theta)$ is convex but not necessarily strongly convex. This implies the loss landscape may contain multiple minimizers rather than a unique global minimum. Consequently, we characterize convergence in terms of $\hat{U}_g(\pi_{\theta^{\mathrm{opt}}}) - \hat{U}_g(\pi_{\bar{\theta}_T})$ (instead of $\|\theta_t - \theta_n^{\mathrm{opt}}\|$). Here, $\theta^{\mathrm{opt}} \in \arg\max_\theta \hat{U}_g(\pi_\theta)$ is an optimal parameter and $\bar{\theta}_T$ is the average of the SGA iterates.

**Proposition D.4.** *Let $\theta^{opt} \in \arg\max_\theta \hat{U}_g(\pi_\theta)$ be an optimal parameter. If the learning rate satisfies $0 < \eta \leq \frac{1}{4\bar{L}}$, then by (Garrigos & Gower, 2023) (Theorem 6.9), the iterates of mini-batch SGA satisfy:*

$$\mathbb{E}\left[\hat{U}_g(\pi_{\theta^{opt}}) - \hat{U}_g(\pi_{\bar{\theta}_T})\right] \leq \frac{\|\theta_0 - \theta^{opt}\|^2}{\eta T} + \frac{8\eta G_{\max}^2 H^2}{b}$$

*where $\bar{\theta}_T$ is the average of the iterates.*

Proposition D.4 highlights the trade-off inherent to constant step-size SGA: a larger $\eta$ accelerates the decay of the initial error (first term) but increases the asymptotic noise floor (second term). For a fixed horizon $T$, one can recover a convergence rate of $\mathcal{O}(1/\sqrt{T})$ by setting $\eta \propto 1/\sqrt{T}$, which balances both terms.

## D.3. PWLL with $\ell_2$ regularization

We now move to the $\ell_2$-regularized case where the PWLL objective is strongly concave (Proposition 3.1). Precisely, we consider the regularized objective $\tilde{U}^\lambda(\theta) = \hat{U}_g(\pi_\theta) - \frac{\lambda}{2}\|\theta\|^2$.

Strong convexity implies the existence of a unique global minimizer. This allows us to guarantee convergence of the parameters $\theta_t$ themselves, which is a stronger condition than value convergence.

**Proposition D.5.** *Let $\theta_{n,\lambda}^{opt} = \arg\max_\theta \tilde{U}^\lambda(\theta)$ be the unique optimal parameter. If the learning rate satisfies $0 < \eta \leq \frac{1}{2(G_{\max}H^2+\lambda)}$, then by (Garrigos & Gower, 2023) (Theorem 6.12):*

$$\mathbb{E}\left[\|\theta_t - \theta_{n,\lambda}^{opt}\|^2\right] \leq (1-\eta\lambda)^t\|\theta_0 - \theta_{n,\lambda}^{opt}\|^2 + \frac{8\eta G_{\max}^2 H^2}{\lambda b}$$

The regularized case demonstrates a convergence rate that is significantly faster than the rate of the unregularized case.

# E. Additional Experiments

## E.1. Detailed Experimental Setting

**Experimental Setting.**

*Table 1.* Statistics of Post Processed Datasets

| Dataset | Num. of actions | Num. of samples |
|---------|-----------------|-----------------|
| MovieLens | $60,000$ | $132,744$ |
| Twitch | $200,000$ | $400,000$ |
| GoodReads | $1,000,000$ | $400,000$ |

Our experimental setup is designed to study the behavior of the different policy learning paradigms in large action spaces. To this end, we use three large action spaces collaborative filtering datasets: Movielens (Lam & Herlocker, 2016), Twitch (Rappaz et al., 2021) and GoodReads (Wan et al., 2019) that are preprocessed to obtain a user-item interaction matrix. We follow the exact procedure of Sakhi et al. (2023b) to pre-process the datasets. The statistics of the obtained datasets are described in Table 1. For each user, we keep half of its history as the context $x$, and use the other half of the history as the products with positive reward, which aligns the learned policies to recommend new and relevant items. We direct the interested readers to Sakhi et al. (2023b) for a detailed description of the experimental setup.

The large action space scenario restricts the policies used to the inner product parametrization (Aouali et al., 2022). This parametrization is essential to leverage Maximum Inner Product Search algorithms (Shrivastava & Li, 2014) for fast query response. In particular, we adopt policies of the following form:

$$\pi_\theta(a|x) \propto \exp(\langle \phi_\Gamma(x), \beta_a \rangle),$$

with the learnable parameter $\theta = [\Gamma, \beta]$, $\phi_\Gamma : \mathcal{X} \to \mathbb{R}^\ell$ defines the context embedding function in $\mathbb{R}^\ell$ and $\beta$ the actions embeddings of size $K \times \ell$. In all our experiments and unless it is explicitly stated, we follow the procedure of (Sakhi et al., 2023c) to define our policies. We start by extracting action embeddings $\beta_0$ using an SVD decomposition of the user-item matrix. These embeddings help us define the context embedding function $\phi_\Gamma$ and our logging policy $\pi_0$. $\phi_0$ is set to the average embeddings of the observed actions in the contexts and is fixed for the logging policy $\pi_0$. Using the SVD action embeddings $\beta_0$, we define our logging policy $\pi_0$ as:

$$\pi_0(a|x) \propto \exp\left(\frac{1}{t}\langle \phi_0(x), \beta_{0,a} \rangle\right) \mathbb{I}\left[a \in \text{TOP}^{k_0}(x)\right],$$

with $t$ the temperature of the logging policy, and $k_0$ defines the support of the logging policy, concentrating on the top $k_0$ actions with: $\text{TOP}^{k_0}(x) = \text{argsort}_{a_1,\cdots,a_{k_0}}\langle \phi_0(x), \beta_{0,a} \rangle$.

If not explicitly stated, $k_0$ is set to 100 and the temperature at $t = 1$ in all experiments. This policy is used to collect the offline dataset $\mathcal{D}_n = \{X_i, A_i, R_i\}_{i \in [n]}$ on which all trainings are conducted. For each $i \in [n]$ in the processed dataset, $X_i$ is the user history, $A_i$ is the action played by the logging policy $\pi_0(\cdot|X_i)$ and $R_i = \mathbb{1}[A_i \in H_i]$ the observed reward, which is if the action played is in the hidden items of user $i$.

**Trained Policies Parameterizations.** We adopt two parameterizations of the trained policies. The first one is a **heavyweight** parametrization, and focuses on learning the embeddings of the actions $\beta$ (be it $\mathcal{A}$ of size $K$ or $\mathcal{C}$ of size $|\mathcal{C}|$), meaning that $\theta$ in this case is $\beta$. For action-level policies, this gives $\beta \in \mathbb{R}^{K \times \ell}$ and for any $x$:

$$\pi_\beta(a|x) = \frac{\exp(\langle \phi_0(x), \beta_a \rangle)}{\sum_{a' \in \mathcal{A}_{\text{eff}}(x)} \exp(\langle \phi_0(x), \beta_{a'} \rangle)},$$

with $\mathcal{A}_{\text{eff}}(x) \subset \mathcal{A}$, which depends on the choice of the practitioner, for example $\mathcal{A}_{\text{eff}}(x) = S_0(x)$, the support of $\pi_0$ for context $x$ when we optimize IPS objectives. For cluster-level policies, this gives a $\beta \in \mathbb{R}^{|\mathcal{C}| \times \ell}$ and for any $x$:

$$\pi_\beta(c|x) = \frac{\exp(\langle \phi_0(x), \beta_c \rangle)}{\sum_{c' \in \mathcal{C}} \exp(\langle \phi_0(x), \beta_{c'} \rangle)}.$$

This is used by default if nothing is explicitly stated.

We also define a **lightweight** parametrization, where only a small projection $W \in \mathbb{R}^{\ell \times \ell}$ is learned, giving in action level policies:

$$\pi_W(a|x) = \frac{\exp(\langle \phi_0(x)W, \beta_{a,0} \rangle)}{\sum_{a' \in \mathcal{A}_{\text{eff}}(x)} \exp(\langle \phi_0(x)W, \beta_{a',0} \rangle)},$$

using $\beta_0$, the embeddings of $\pi_0$. For cluster level policies, we first define $\bar{\beta}_0 \in \mathbb{R}^{|\mathcal{C}| \times \ell}$ with $\bar{\beta}_{0,c} = \frac{1}{|c|} \sum_{a \in c} \beta_{0,a}$, and use it to define the cluster level policy:

$$\pi_W(c|x) = \frac{\exp(\langle \phi_0(x)W, \bar{\beta}_{c,0} \rangle)}{\sum_{c' \in \mathcal{C}} \exp(\langle \phi_0(x)W, \bar{\beta}_{c',0} \rangle)}.$$

**Reward Model.** The reward model used $\hat{r}$ is learned using regularized linear regression the collected interaction data, with $\hat{r}(x,a) = \langle \phi(x), \theta_a \rangle$.

**Clustering and $\epsilon$ used.** We use the embeddings $\beta_0$, combined with K-means clustering to find our clusters. The number of clusters is set to 2000 for all datasets and experiments. For PC, the $\ell_2$ threshold $\epsilon$ is set to 0.1.

### E.2. Additional results

**Benefits of objective-aware parametrization.** Figure 5 shows the effect of objective-aware policy parameterizations for two different objectives and three large action space datasets.

**Effect of Objective-Aware Parametrization on Performance**

*Figure 5.* The effect of objective-aware parametrization for IPS and cLPI on three large-scale datasets

**MSE progress during training.** Figures 6a to 6c show the progress of the MSE over 10 epochs on all three datasets. Several methods are excluded from the figure, as their high MSE values would distort the scale and obscure the comparison.

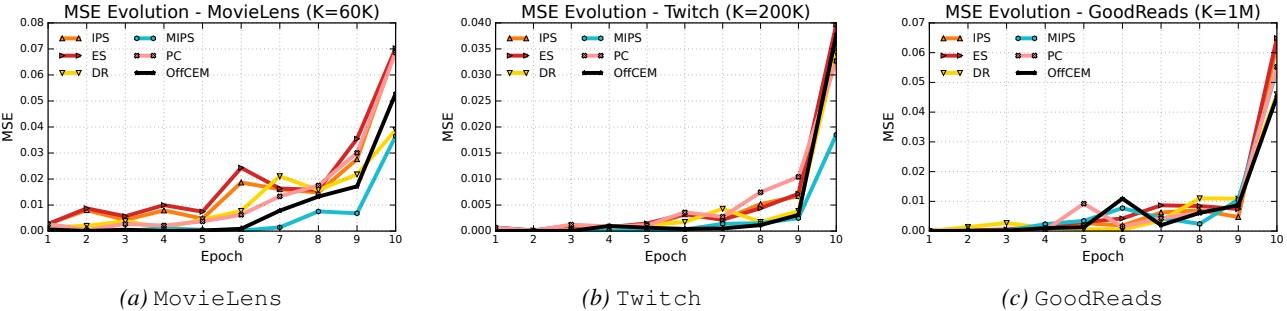

*(a)* MovieLens        *(b)* Twitch        *(c)* GoodReads

*Figure 6.* MSE progression over 10 epochs across datasets.

### E.3. Results Averaging Different Seeds

In this experiment, we analyze the reward evolution of representative PWLL and IPS-based methods on the three considered datasets. We compare two distinct optimization configurations: (i) a standard off-the-shelf Adam optimizer, and (ii) a carefully tuned setup using Adam with an optimized batch size and a one-cycle learning-rate scheduler. This comparison enables us to isolate the effect of optimization on stability and convergence. Each method is evaluated over 5 random seeds, and we report the mean reward along with a shaded standard deviation region to visualize sensitivity to optimization randomness.

In Figure 7, across all datasets and optimization settings, we observe that IPS-based methods (cIPS, IX, and even POTEC) not only reach inferior performance but also suffer from considerably higher variance. Their uncertainty bands are significantly wider, indicating unstable optimization. In contrast, PWLL-based methods exhibit near-invisible variance bands, with standard deviations roughly an order of magnitude smaller on average.

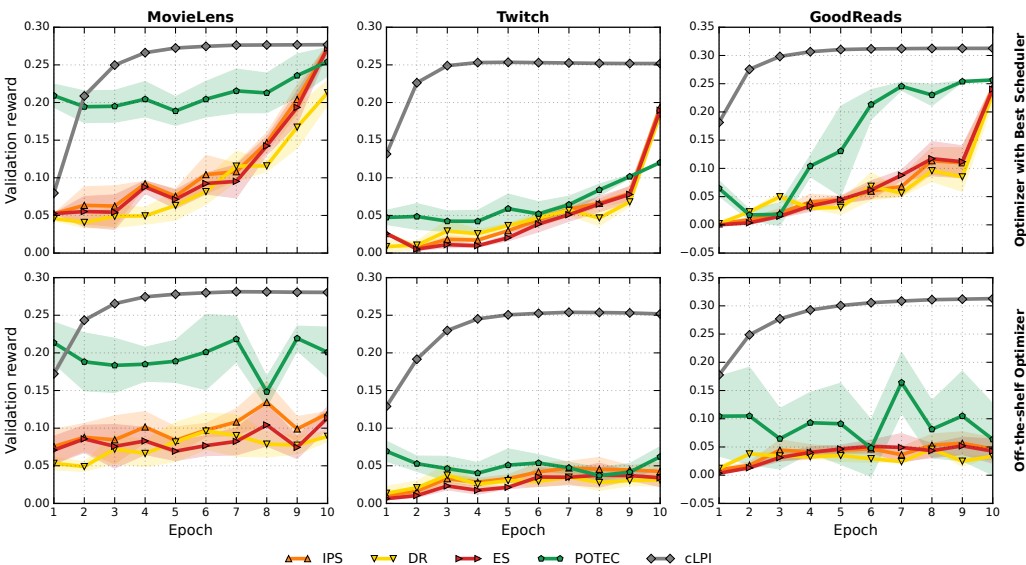

*Figure 7.* cLPI vs IPS-Based methods: Evolution of rewards averaged over 5 different seeds. cLPI is more stable to optimize and reaches better policies.

Finally, in Figure 8, we observe that adopting an Objective-Aware parametrization yields further performance and stability improvements. For example, cIPS with Objective-Aware parametrization surpasses cIPS while maintaining lower variability, and cLPI in its Objective-Aware form consistently achieves the best overall performance. These results

demonstrate that the combination of PWLL objectives and clever parametrization leads to more robust and more effective learned policies, while being very simple to implement.

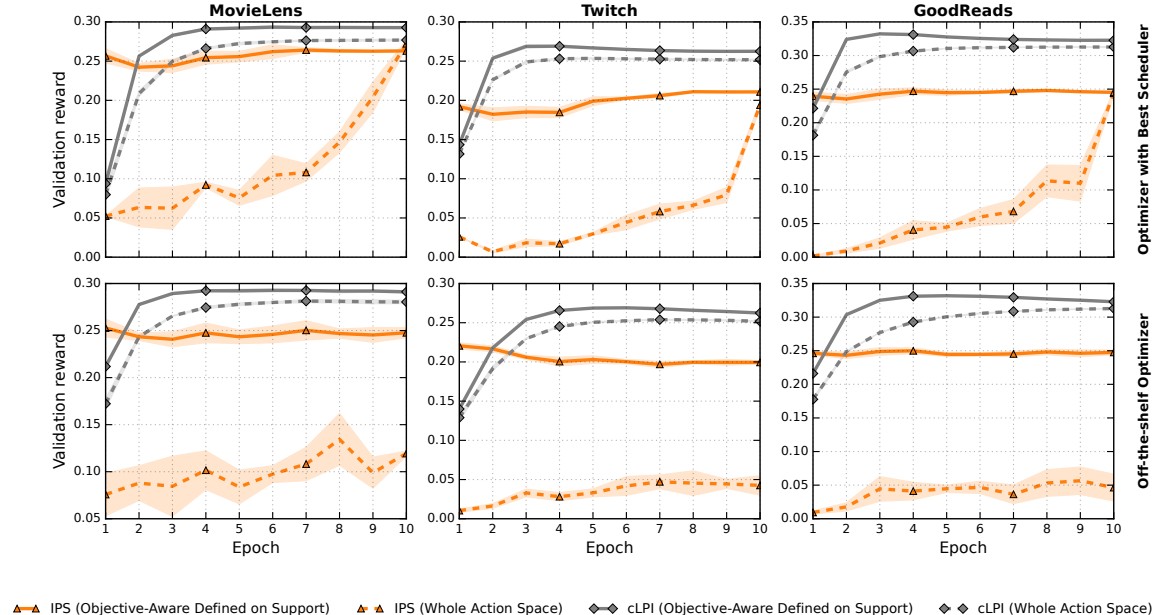

*Figure 8.* Objective Aware Parametrization: Evolution of rewards averaged over 5 different seeds. Objective Aware Parametrization stabilizes and improves performance for PWLL and IPS methods.

### E.4. Ablation - Sensitivity to Reward Noise

In the original evaluation setup (see Appendix E), the observed reward is deterministic; for user $i$, we have $R_i = \mathbb{1}[A_i \in H_i]$, meaning that a positive reward is returned only when the selected action belongs to the user's hidden set $H_i$. In this section, we investigate robustness to reward noise by introducing stochasticity in the form:

$$R_i \sim \mathbb{1}[A_i \in H_i]\,(1 - B(\epsilon)) + B(\epsilon)\,s,$$

where $B(\epsilon)$ is a Bernoulli random variable with parameter $\epsilon$, and $s \in [0, 1]$ is a shift. This results in noisy rewards supported on $[0, 1]$. Note that any reward scaling can be normalized to this range via $R/R_{\max}$ when $R_{\max} > 1$.

We evaluate six configurations defined by noise parameters $\epsilon \in \{0.1, 0.2, 0.3\}$ and reward shifts $s \in \{0, 0.5\}$. All methods are trained using the best-performing optimization schedule (one-cycle) to isolate the effect of noise. Results are reported in Figure 9.

We observe that increasing the noise level when $s = 0$ consistently harms all methods, as expected from a more stochastic reward signal. In contrast, when $s = 0.5$, higher noise tends to increase the overall reward level, since the shift raises the baseline reward. Across all noise–shift conditions, PWLL-based objectives maintain a clear advantage over IPS-based methods. When $s = 0$, RegKL and cLPI perform similarly, confirming that both benefit from the logarithmic reparameterization. However, as both noise and shift increase, RegKL begins to outperform cLPI, suggesting that, with an appropriately chosen regularization weight $\beta$, RegKL remains highly competitive even under high stochasticity reward.

**Conclusion.** PWLL methods demonstrate robustness to reward noise, leading to improved stability and performance compared to traditional IPS-based objectives, even in challenging noise regimes.

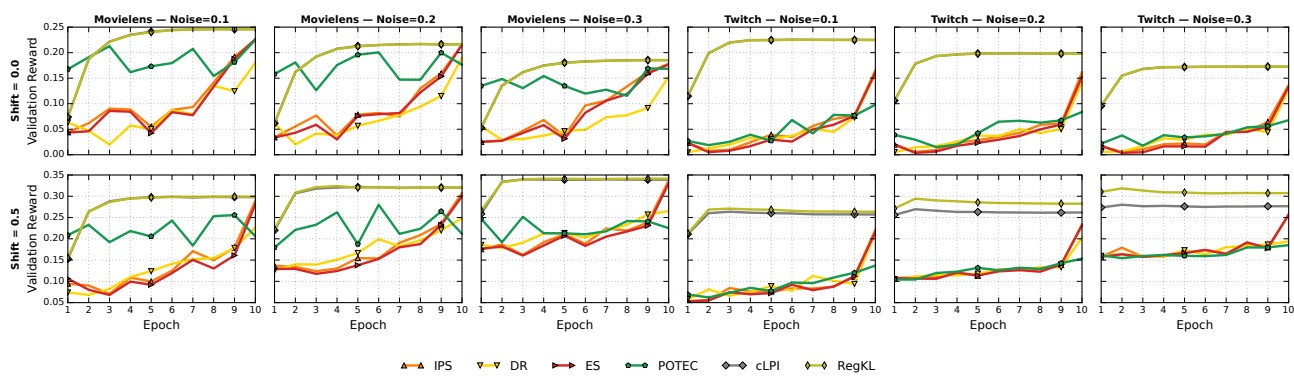

*Figure 9.* Ablation - Sensitivity to reward noise

### E.5. Ablation Study on Hyperparameters and Log Transform

In this section, we evaluate the impact of hyperparameter choices on cIPS, cLPI, RegKL, and RegKL-LIN (the non-logarithmic variant of RegKL). All methods are run using the best-performing optimization configuration (optimizer + learning rate scheduler), ensuring that differences are driven solely by hyperparameter values and by whether the policy transformation is linear or logarithmic. The results are shown in Figure 10.

- **cIPS** consistently fails to reach competitive performance across all values of $\tau$, especially in large action spaces. Its PWLL counterpart, cLPI, dominates for every $\tau$, converging faster and achieving superior results.

- For the KL-based objectives, we restrict to $\beta \geq 0.1$ in order to avoid numerical instability from the exponential term $(\exp(1/\beta) > 2 \cdot 10^5$ for $\beta < 0.1)$. The same trend is observed: the PWLL variant (RegKL) reliably outperforms its linear analogue (RegKL-LIN) across all $\beta$, exhibiting more stable training dynamics, faster convergence and better performance.

**PWLL dominates.** Across both objective families, replacing linear weights with *log-transformed* policy weights (PWLL) consistently provides **greater robustness to hyperparameters**, **faster optimization**, and **higher final performance**, even more in challenging large-action-space settings.

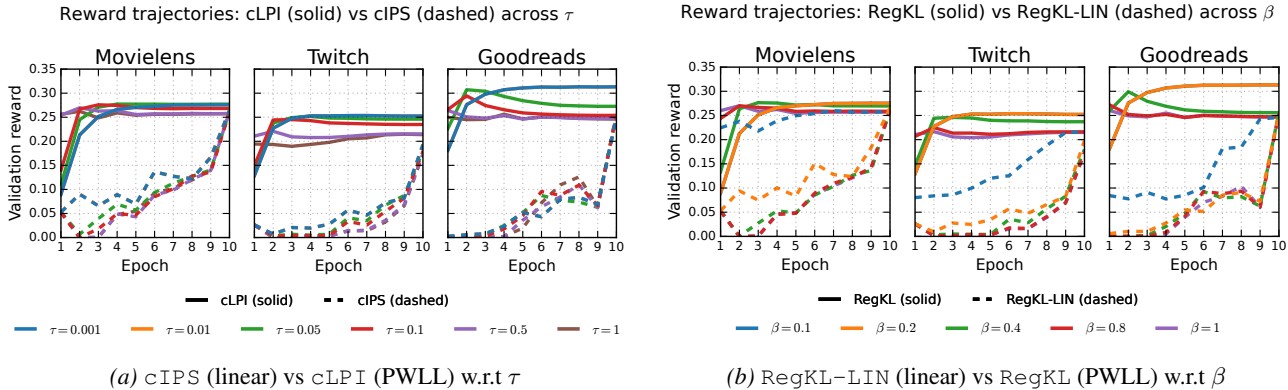

*(a)* cIPS (linear) vs cLPI (PWLL) w.r.t $\tau$

*(b)* RegKL-LIN (linear) vs RegKL (PWLL) w.r.t $\beta$

*Figure 10.* Ablation Study on hyper-parameters and Log Transform

### E.6. Ablation: PWLL in Smaller Action Spaces

We have shown that PWLL provides a more benign optimization landscape and yields stronger policies than IPS-based objectives in large action spaces. Here, we examine whether these benefits also extend to smaller action spaces. We construct a reduced version of Movielens by subsampling the action space to $K \in \{100, 500, 1000, 5000\}$ items.

Figure 11 reports performance across varying $K$ for `cIPS` (linear) and its PWLL-enhanced counterpart `cLPI` (log). In small action space settings ($K \leq 500$), `cIPS` converges faster than `cLPI`, but `cLPI` identifies a better maximum by the end of the 10 epochs. For medium action spaces ($K \geq 500$), `cLPI` consistently outperforms `cIPS`, converging faster and identifying a better maximum. These results indicate that the optimization advantages of PWLL can still be beneficial in medium sized action space settings.

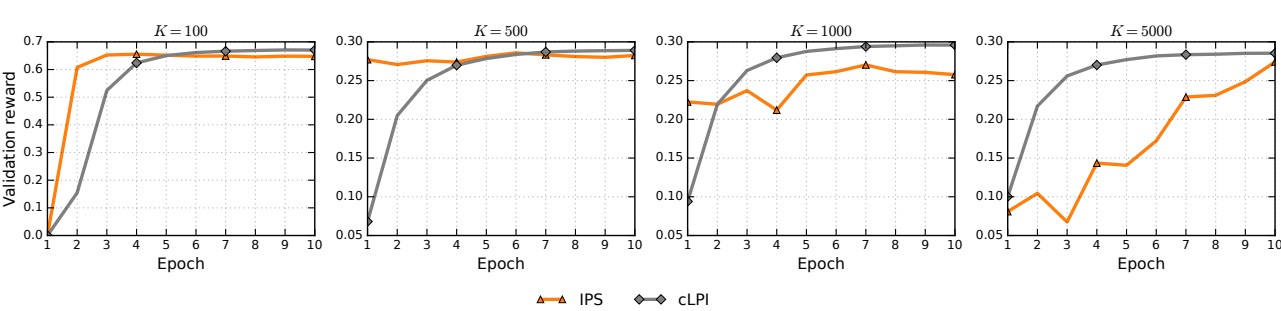

*Figure 11.* PWLL (`cLPI`) vs IPS-based (`IPS`) in smaller action spaces.

### E.7. Ablation - Sensitivity to the number of clusters

In this study, we compare our simple PWLL objective `cLPI` against `MIPS` and `POTEC`, two more complex IPS-based methods specifically designed for large action spaces. These baselines rely on a clustering function to reduce variance, and `POTEC` additionally leverages a reward model $\hat{r}$. We examine how the number of clusters affects their optimization performance. Figure 12 reports the results.

`POTEC` generally outperforms `MIPS` for all numbers of clusters. However, both methods exhibit optimization instability across settings. While `POTEC` can occasionally match the final performance of `cLPI` on Movielens for a carefully selected number of clusters (1000), it consistently falls short on Twitch regardless of the cluster configuration.

**Conclusion.** These findings demonstrate that focusing on optimization properties pays off: despite its simplicity, the PWLL objective `cLPI` can consistently outperform intricate IPS-based approaches tailored to large action spaces, even with the best finetuning.

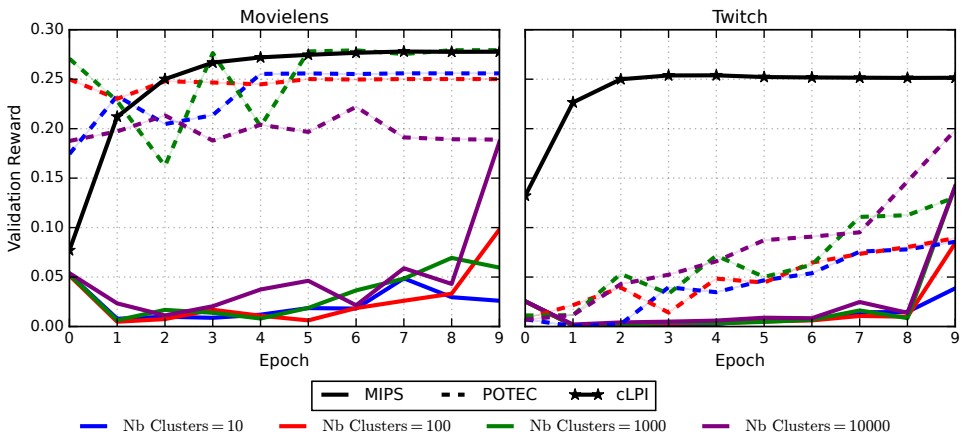

*Figure 12.* PWLL (`cLPI`) vs `POTEC` and `MIPS`, changing the number of clusters.

### E.8. Ablation - Different Logging Supports

We conduct experiments to quantify how increasing or restricting the support of the logging policy affects policy learning, comparing PWLL and IPS-based methods. Figure 13 compiles the results and show that PWLL is still better than IPS-based

approaches for different logging support sizes.

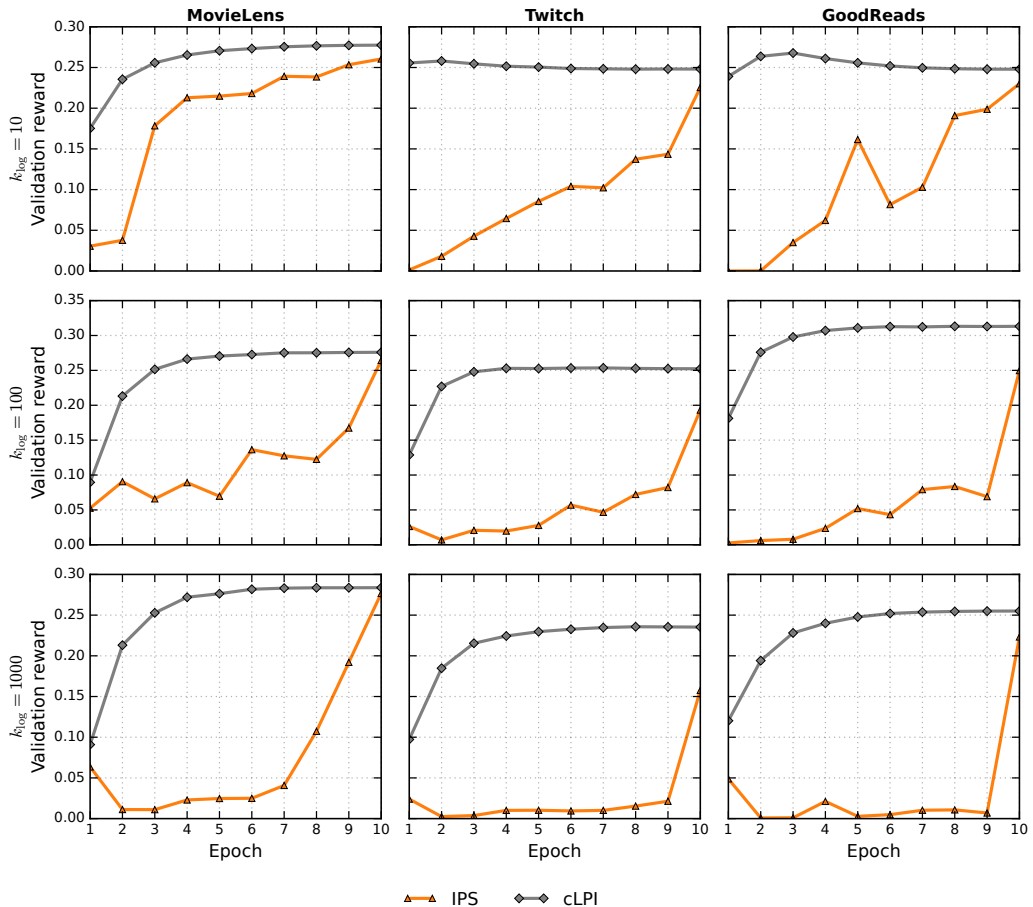

*Figure 13.* PWLL vs IPS: Different Logging Support sizes $k_{\log}$

### E.9. Pessimism Does not Solve Optimization Problems

Pessimism in the face of uncertainty Jin et al. (2021) is motivated through a pure statistical learning rationale and is used to provide better statistical guarantees and more controlled excess risk. In the context of OPL, pessimistic strategies are derived combining concentration bounds with class complexity measures, be it VC dimension (Swaminathan & Joachims, 2015a) or PAC-Bayesian tools (London & Sandler, 2019; Aouali et al., 2023; Sakhi et al., 2024). For example, in its PAC-Bayesian formulation, the pessimistic objectives are all written in the following form:

$$\arg\max_{\pi_\theta} \quad \hat{V}(\pi_\theta) - \frac{\lambda}{n}||\theta - \theta_0||_2^2 \, ,$$

Adding an $\ell_2$ regularization term that pulls the parameters $\theta$ towards the behavior policy parameters $\theta_0$ (defining $\pi_0$) induces pessimism by encouraging the learned policy to stay close to $\pi_0$ in parameter space. However, the optimization landscape of this objective becomes concave only when the regularization weight $\lambda$ is sufficiently large for the $\ell_2$ term to dominate. In that regime, the objective is indeed easier to optimize, but becomes overly conservative, yielding policies that remain too close to $\pi_0$ and under-exploit potential improvements. Figure 14 confirms this empirically: pessimistic approaches, whether based on Sample Variance Penalisation (SVP) (Swaminathan & Joachims, 2015a), PAC-Bayesian learning with clipped IPS (London & Sandler, 2019), Exponential Smoothing (Aouali et al., 2023), or Logarithmic Smoothing (Sakhi et al., 2024), fail to outperform cLPI for any value of $\lambda$.

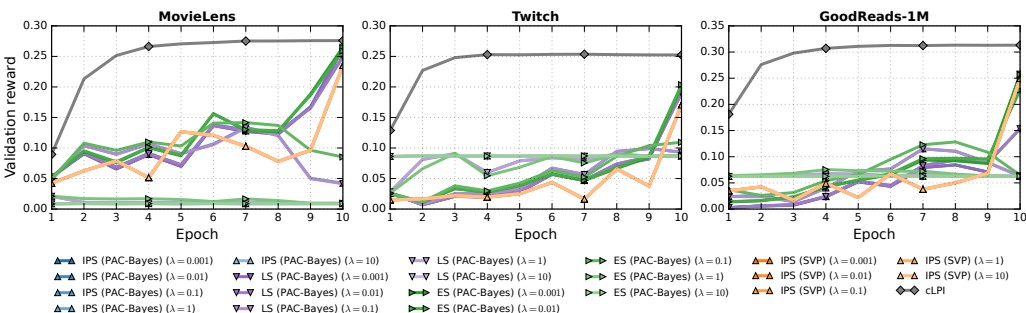

*Figure 14.* cLPI outperforms the pessimistic approaches. Some methods do not appear in the plot because their curves overlap.

