# OpenReview forum: "Off-Policy Learning in Large Action Spaces: Optimization Matters More Than Estimation"
_ICML.cc/2026/Conference — ICML 2026 regular_

### Official Review · Reviewer_ccm2 · 2026-03-09

**Soundness:** 2
**Presentation:** 2
**Significance:** 2
**Originality:** 2
**Overall Recommendation:** 3
**Confidence:** 3

**Summary:**

This paper studies off-policy evaluation and off-policy learning when the number of actions is large. It provides theoretical evidence on optimization challenges faced by IPS-based estimators under large action space, and proposes objective-aware parametrization to mitigate these issues. In addition, the paper introduces a class of policy-weighted log-likelihood (PWLL) objectives that trade value estimation accuracy for concave optimization landscape. Empirical experiments are conducted to illustrate the optimization challenges in off-policy learning.

**Compliance With Llm Reviewing Policy:**

Affirmed.

**Final Justification:**

I thank the authors for their detailed responses. While my concerns regarding the overall contribution are not fully resolved, I acknowledge that the re-examination perspective offered by the paper is useful and have adjusted my score accordingly.

**Key Questions For Authors:**

see above

**Limitations:**

Yes

**Strengths And Weaknesses:**

Strength:
* The problem of understanding the optimization properties of off-policy learning objectives is important. In large action spaces, IPS-based estimators can induce challenging optimization landscapes, making principled approaches to improving optimization stability valuable.

Weakness:
* Although the paper identifies an important problem, the presentation of the proposed solution does not appear to be sufficiently complete. For example, in Section 2.4 the authors note that the key is to find the “sweet spot” for $K_{\text{eff}}$, but provide only limited discussion on how this value should be determined in practice. It would also be helpful to include theoretical justification for Claims 2.3 and 2.4.

* In addition, the current presentation reads somewhat like a summary of existing IPS-based methods rather than a fully developed contribution for a top venue such as ICML. Strengthening the technical development and more clearly articulating the novelty relative to prior work would further improve the paper.

---

> ### Author Rebuttal · Authors · 2026-03-30
>
> Thank you for your time and feedback.
>
> On the “sweet spot” for $A_{\mathrm{eff}}$, the choice is in fact **made explicit in Claims 2.3 and 2.4**. In summary, when a method does not assume any clustering structure, $A_{\mathrm{eff}}$ should correspond to the support size of the logging policy, that is, the number of actions assigned non-zero probability by the logger. When a clustering structure is used, $A_{\mathrm{eff}}$ should instead correspond to the number of clusters, together with a decomposed policy class such as the one used by POTEC. These two cases cover all the methods studied in the paper, and our experiments validate that these choices perform well in practice.
>
> We also want to clarify the role of Claims 2.3 and 2.4. They are intentionally stated as **claims**, not theorems. **Their purpose is to provide design guidance derived from the oracle-policy analysis**, which is one of the paper’s theoretical contributions, and then supported empirically in the experiments. A full formal proof of these design implications is not conventional and would indeed be very interesting, but it is technically challenging and beyond the scope of the current paper.
>
> Regarding the presentation style, our contribution is precisely to show that IPS-based methods degrade in large action spaces because of optimization difficulties, and to introduce PWLL objectives whose oracle policies remain close in spirit to IPS-type solutions while being much easier to optimize. Making this case necessarily requires revisiting existing IPS-based methods through an optimization lens. This is why the paper does not merely list prior methods as part of a survey; rather, it analyzes them, derives their oracle policies, characterizes their failure modes in large action spaces, and then develops corresponding remedies. In that sense, this re-examination is itself a core contribution of the paper: we do not just summarize existing methods, we explain where they break, why they break, and how to fix the issue.
>
> We hope these clarifications address your main concerns. If the rebuttal resolves your concerns, we kindly ask you to reconsider your score.

---

> > ### Author Rebuttal · Reviewer_ccm2 · 2026-04-04
> >
> > I thank the authors for their detailed responses. While my concerns regarding the overall contribution are not fully resolved, I acknowledge that the re-examination perspective offered by the paper is useful and have adjusted my score accordingly.

---

> > > ### Author Response · Authors · 2026-04-04
> > >
> > > Thank you for the follow-up and for reconsidering the paper. We believe the remaining concern can be addressed now without significant update to the paper by clarifying the presentation of the contributions, and that this is addressable in the revised version as follows.
> > >
> > > Our **first contribution** is to show theoretically that PWLL objectives preserve a **learning inductive bias** close in spirit to IPS objectives while being substantially easier to optimize. To make this precise, we re-examine IPS-based objectives through their learning (rather than estimation) inductive bias, **captured by oracle policies**. This is not a survey step, but a core analysis: we derive the oracle policies of these methods and expose their implicit learning bias. We then show that these IPS-based objectives are hard to optimize in large action spaces (Props. 2.1–2.2). In contrast, the paper shows that PWLL objectives retain learning biases close in spirit to their IPS-based counterparts while being much easier to optimize, with concavity for linear-softmax policies (Prop. 3.1) and global convergence guarantees under gradient ascent (Thms. D.4–D.5). This establishes our first contribution.
> > >
> > > Our **second contribution** is to show that the optimization difficulties of IPS objectives scale with the **effective action space** $\mathcal{A}\_{\mathrm{eff}}$ (Props. 2.1–2.2), which itself depends on the policy parametrization, and that $\mathcal{A}\_{\mathrm{eff}}$  can be chosen to improve optimization without introducing additional learning bias. Concretely, a smaller $\mathcal{A}\_{\mathrm{eff}}$  makes optimization easier, but can also introduce bias if the parametrization becomes too restrictive. Our theoretical oracle-policy analysis gives the right way to resolve this tradeoff: choose the smallest parametrization that still contains the oracle policy, so that optimization improves without changing the learning target. This is precisely the role of **objective-aware parametrization**. In practice, this yields concrete rules that are motivated by the theory, summarized in Claims 2.3–2.4 as practical guidelines, and Section 4 and Appendix E show empirically that they lead to clearly much better learned policies.
> > >
> > > So, to summarize, the contribution is not simply to list existing IPS-based objectives, but to:
> > >
> > > (1) derive the oracle policies (inductive learning bias) of IPS objectives,
> > >
> > > (2) reveal how their optimization difficulty scales in large action spaces,
> > >
> > > (3) introduce PWLL objectives with similar oracle policies but much better optimization properties,
> > >
> > > (4) show that optimization issues of IPS objectives scale with $\mathcal{A}\_{\mathrm{eff}}$,
> > >
> > > (5) use the theory to choose the smallest $\mathcal{A}\_{\mathrm{eff}}$ that still preserves the oracle policy, yielding **objective-aware parametrization**, and
> > >
> > > (6) turn these insights into practical design rules that improve learning in practice (Claims).
> > >
> > > We are grateful for your feedback, as it helped us realize that this contribution flow should be stated much more explicitly. We will make this clearer in the revised version, and we hope this now clarifies and addresses your concern.

---

### Official Review · Reviewer_UM63 · 2026-03-11

**Soundness:** 2
**Presentation:** 3
**Significance:** 3
**Originality:** 3
**Overall Recommendation:** 5
**Confidence:** 3

**Summary:**

The authors propose a shift of perspective from estimation-centered off-policy evaluation for contextual bandits to optimization-based learning. While importance propensity scoring methods yield good approximations for estimating the true policy value functions, these methods often yield challenging non-convex optimization landscapes with exponentially many local minima requiring linearly many steps to escape wrt. the number of actions. Motivated by linear-softmax policy parametrization, policy-weighted-log-likelihood provides a powerful alternative with a concave landscape, which the authors underpin with an illustrative toy example. On benchmarks of different scales, the authors ablate the sensitivity of estimation-centered vs optimization-centered off-policy learning and conclude that for large action spaces, an optimization perspective is favorable.

**Compliance With Llm Reviewing Policy:**

Affirmed.

**Final Justification:**

I appreciate the authors’ detailed responses and clarifications, which address my main concerns satisfactorily. Based on this, I am raising my score to 5.

I would like to reiterate that the theoretical limitation highlighted in Remark 3.3, namely the absence of a concavity guarantee beyond log linear softmax policies, is a genuine and non trivial constraint on the scope of the contribution. This is not a minor caveat, and I appreciate that the authors acknowledge it clearly rather than downplaying it. At the same time, the alignment between the theoretical setting based on inner product linear softmax parameterizations and the experimental setup is reasonable, and the large scale benchmarks provide convincing empirical support for the practical relevance of the proposed optimization perspective. The ablation studies are thorough, and the work appears technically sound overall.

I also appreciate the authors’ commitment to include additional ablation results in the main text and to further clarify the limitation and its implications in the revision.

**Key Questions For Authors:**

A clearer connection between existing methods on the RL site and the proposed PWLL methods would be very appreciated. In particular, since also in RL the trade-off between optimization and estimation exists and leads to the paradigm of advantage-weighted methods.

**Limitations:**

Yes

**Strengths And Weaknesses:**

**Strengths:**

-- The paper is well-written and clear. The literature review and presentation of methods are comprehensive, and the mathematical notations are rigorous.

-- The change of perspective from estimation to optimization is well motivated and gives strong insights, especially for practitioners. The work provides clear guidelines/tips for large-scale off-policy learning.

-- The authors provide nice toy examples to demonstrate the differences in the optimization landscape and correctly emphasize the optimization challenges one faces when dealing with IPS-based objectives.

**Weaknesses:**

-- Remark 3.3 correctly identifies the main (strong) limitation of this work, i.e., the missing concavity guarantee beyond log-linear-softmax policies in practice. This is a key assumption and a central cornerstone for the motivation of PWLL objectives.

-- Under the previously mentioned weakness, the main contribution of this paper is mostly a change of perspective and very comprehensive ablation studies. Personally, I would find it very useful to put more figures/results in the main part, which would also be a good opportunity to show the well-crafted ablation studies in the appendix.

-- The authors correctly mention the connection to behavioral cloning in RL, but unfortunately, this road is not further pursued. A closer comparison and positioning of the work would significantly strengthen this paper.

**Minor comments:**

-- It is slightly confusing that in Figure 2 and Figure 3 the order of the LR schedulers are not the same. It could be beneficial to adjust this.

---

> ### Author Rebuttal · Authors · 2026-03-30
>
> Thank you for the careful reading and for emphasizing the practical importance of the optimization perspective. We agree that the concavity guarantee is limited to linear-softmax policies, and that this is an important caveat. At the same time, all of our experiments use inner-product linear-softmax policies, so the theoretical setting is well aligned with the empirical one. We also note that these simple policies already perform very strongly in our large-scale benchmarks. This limitation is acknowledged in the paper, and we will make it even more explicit in the revision and it is an interesting direction for future work .
>
> Regarding the request to move more results from the appendix into the main paper, we fully agree. This was primarily due to page constraints. In the camera-ready version, we will use the additional space to surface more of the ablations in the main text.
>
> On the connection to offline RL and behavior-cloning-style methods, our intent is not to present PWLL as a novel update rule, and we state this in the paper. Rather, our goal is to show that the optimization-versus-estimation tradeoff is very important in large-action contextual bandits, where IPS-based off-policy learning remains the dominant approach.
>
> Finally, we appreciate the presentation suggestions. The inconsistency in the LR scheduler ordering is a great catch, and we will fix it in the revision.
>
> We sincerely appreciate your time and positive feedback. We hope the additional clarifications fully resolve your concerns, and we would greatly appreciate your consideration in raising your support for our paper.

---

> > ### Author Rebuttal · Reviewer_UM63 · 2026-04-05
> >
> > Thank you for the thorough and thoughtful rebuttal. I appreciate the authors’ detailed responses and clarifications, which address my main concerns satisfactorily. Based on this, I am raising my score to 5.
> >
> > I would like to reiterate that the theoretical limitation highlighted in Remark 3.3, namely the absence of a concavity guarantee beyond log linear softmax policies, is a genuine and non trivial constraint on the scope of the contribution. This is not a minor caveat, and I appreciate that the authors acknowledge it clearly rather than downplaying it. At the same time, the alignment between the theoretical setting based on inner product linear softmax parameterizations and the experimental setup is reasonable, and the large scale benchmarks provide convincing empirical support for the practical relevance of the proposed optimization perspective. The ablation studies are thorough, and the work appears technically sound overall.
> >
> > I also appreciate the authors’ commitment to include additional ablation results in the main text and to further clarify the limitation and its implications in the revision. I look forward to the camera ready version.

---

> > > ### Author Response · Authors · 2026-04-07
> > >
> > > We sincerely thank the reviewer for the thoughtful follow-up and positive assessment of our work. We are very grateful that our rebuttal addressed your main concerns satisfactorily.
> > >
> > > As a side note, we noticed that the official form still displays the previous score of 4. We thought it might be useful to bring this to your attention only in case the intended update has not yet registered in the system.
> > >
> > > We sincerely appreciate your time and the rigorous feedback you have provided throughout this process, and we will definitely incorporate all your remarks in the camera-ready version.

---

### Official Review · Reviewer_uwaU · 2026-03-13

**Soundness:** 3
**Presentation:** 3
**Significance:** 2
**Originality:** 2
**Overall Recommendation:** 3
**Confidence:** 4

**Summary:**

This paper investigate the optimization in surrogates offline contextual bandit problem by introducing a surrogate objective, and study the performance comparison with IPS type methods.

**Compliance With Llm Reviewing Policy:**

Affirmed.

**Final Justification:**

I view this rebuttal as only a partial resolution: matching the argmax action is weaker than fully matching the policy-level inductive bias.

**Key Questions For Authors:**

see weaknesses

**Limitations:**

yes

**Strengths And Weaknesses:**

Strengths

The writing is good.


Weakness

- This paper over claims its scope. It is titled "off-policy learning" but the setting is much narrower in the offline contextual bandits, where the distribution shifts only appear in treatment and the outcome.
- The theory explains why IPS type objectives are hard to optimize and why the proposed surrogate is better, but doesn't show it will lead to a stronger policy learning.
- The paper is leaning to a experimental study paper, but the experiments are limited to a small number of recommendation type datasets and specific model choices, which is not s strong general claim for optimization matters more than estimation in large action off-policy learning.

---

> ### Author Rebuttal · Authors · 2026-03-30
>
> Thank you for your careful reading and feedback. Regarding the title and scope, we agree that our analysis and experiments are specifically in **offline contextual bandits**. At the same time, the terminology **off-policy learning/evaluation** is standard in the offline contextual bandit literature (see [1–6], among many others), which is why we used that phrasing. We nevertheless agree that the scope can be stated more precisely, and we will make this explicit in the revised abstract.
>
> We also want to clarify that we do **not** claim a theorem stating that PWLL always learns a better policy than IPS-based objectives. Rather, our contribution is **two-fold, both theoretical and empirical**, and we believe it is precisely the combination of these two parts that makes the paper’s message solid. On the theoretical side, we show that IPS-based objectives can exhibit exponentially many local maxima, and that gradient ascent can remain trapped in plateaus for a number of iterations that scales with the action space size. By contrast, PWLL admits global theoretical convergence guarantees under gradient ascent. This establishes that there are regimes in which optimization makes IPS-based learning objectives less favorable than PWLL, even when the former have stronger estimation properties.
>
> On the empirical side, we show that this optimization advantage can indeed translate into better learned policies in practice. Across three public large-scale recommendation datasets with **60k, 200k, and 1M actions**, we observe the same pattern consistently through extensive ablations, including sweeps over batch size and learning-rate schedules, multiple seeds, reward-noise stress tests, smaller-$K$ ablations, and support/cluster sensitivity analyses. In terms of both action-space scale and robustness analysis, this evaluation goes substantially beyond what is typical in this literature. Taken together, the theoretical and empirical results support the paper’s main claim: in large-action offline contextual bandits, optimization can matter more than estimator accuracy for successful policy learning.
>
> On experimental breadth, we chose three public large-scale recommendation benchmarks precisely because this is the regime where large action spaces, inner-product policies, and offline contextual-bandit learning arise most naturally in practice. We also note that multiple reviewers found the empirical evidence convincing, which suggests that, within this target regime, the main message of the paper is well supported. More broadly, as noted above, the evaluation goes beyond typical prior work in this literature in terms of both scale and ablation depth.
>
> We sincerely appreciate your time and feedback. We hope the additional clarifications fully resolve your concerns, and we would greatly appreciate your consideration in raising your support for our paper.
>
> [1] Metelli, A. M., Russo, A., and Restelli, M. Subgaussian and differentiable importance sampling for off-policy evaluation and learning. Advances in Neural Information Processing Systems, 34:8119–8132, 2021.
>
> [2] Aouali, I., Brunel, V.-E., Rohde, D., and Korba, A. Exponential smoothing for off-policy learning. In International Conference on Machine Learning, pp. 984–1017. PMLR, 2023.
>
> [3] Kuzborskij, I., Vernade, C., Gyorgy, A., and Szepesvari, ´ C. Confident off-policy evaluation and selection through self-normalized importance weighting. In International Conference on Artificial Intelligence and Statistics, pp. 640–648. PMLR, 2021.
>
> [4] Saito, Y. and Joachims, T. Off-policy evaluation for large action spaces via embeddings. In Proceedings of the 39th International Conference on Machine Learning, volume 162, pp. 19089–19122. PMLR, 2022.
>
> [5] Jeunen, O. and Goethals, B. Pessimistic reward models for off-policy learning in recommendation. In Fifteenth ACM Conference on Recommender Systems, pp. 63–74, 2021.
>
> [6] Farajtabar, M., Chow, Y., and Ghavamzadeh, M. More robust doubly robust off-policy evaluation. In International Conference on Machine Learning, pp. 1447–1456. PMLR, 2018.

---

> > ### Author Rebuttal · Reviewer_uwaU · 2026-04-03
> >
> > The rebuttal improves the presentation of contribution and makes the empirical messages more precise. However, I do not think it fully isolates optimization from objective-induced bias. Although the rebuttal is helpful and appreciated, it does not significantly change my overall evaluation.

---

> > > ### Author Response · Authors · 2026-04-04
> > >
> > > Thank you for the prompt follow-up and for clarifying your concern. Your mention of **objective-induced bias / inductive bias** was very helpful, as it made the concern much clearer to us and allows us to address it directly.
> > >
> > > To address it clearly, we first make precise what we mean by inductive bias in this setting. The relevant notion here is the **oracle policy**, since we study **learning** objectives and ultimately care about **what policy is learned** and how biased that learned policy is. In that sense, the oracle policy is the right object to examine: it captures the population target of the objective. With this definition in place, we address your concern from two angles.
> > >
> > > Our first point is that **PWLL and IPS-based objectives have very similar (or matching) learning inductive bias while having very different optimization properties**. To make this concrete, we compare **cIPS** (IPS-based objective) and **cLPI** (PWLL-based objective). Their oracle policies exhibit a very similar **clipping-induced bias**. Precisely, the cIPS oracle is
> > >
> > > $$ \pi^*_{\mathrm{cIPS}}(a \mid x) = \mathbf{1} [ a=\arg\max_{a'\in\mathcal A} r(x,a') \frac{\pi_0(a' \mid x)}{\max(\pi\_0(a'\mid x),\tau)}], $$
> > >
> > > while the cLPI oracle is
> > >
> > > $$ \pi^*_{\mathrm{cLPI}}(a\mid x)\propto r(x,a)\frac{\pi_0(a\mid x)}{\max(\pi_0(a\mid x),\tau)}. $$
> > >
> > > Thus, cLPI and cIPS are nearly identical in bias: cIPS is deterministic, while cLPI is stochastic, and this difference disappears under an **argmax** decision rule, where cLPI selects exactly the cIPS oracle action. Their optimization properties, however, are fundamentally different. PWLL is much easier to optimize (Fig. 1): it is concave for linear-softmax policies (Proposition 3.1) and admits global convergence guarantees under gradient ascent (Theorems D.4–D.5), whereas cIPS can still suffer from poor optimization landscapes (Propositions 2.1–2.2). Hence, PWLL-based methods retain a learning bias very close, or matching, their IPS-based counterparts, while gaining substantial optimization tractability, which translates into stronger practical performance (Section 4 and Appendix E).
> > >
> > > Our second point is that our goal was not necessarily to isolate optimization from objective-induced bias, but rather to show that such bias can be a **tool for easier optimization**. This is one of the paper’s central messages: in large action spaces, the right inductive bias can make optimization substantially more benign. This is exactly the logic behind our effective-action-space analysis: by matching the parametrization to the oracle structure, we reduce $K_{\mathrm{eff}}$ and improve trainability. This is also why objective-aware parametrization improves even IPS-based methods (Figure 3).
> > >
> > > Thank you again for this helpful follow-up. We hope this addresses your concern, and we are grateful for your feedback and the time you devoted to reviewing our paper.

---

### Official Review · Reviewer_qhm8 · 2026-03-13

**Soundness:** 3
**Presentation:** 3
**Significance:** 3
**Originality:** 3
**Overall Recommendation:** 5
**Confidence:** 3

**Summary:**

This work studies off-policy learning in large action spaces for contextual bandits. It argues that in the large action number setting, previous popular approaches( e.g., the IPS-based approaches) suffers from optimization issues when taking the policy optimization, including being trapped at a sub-optimal region for a long time and exponentially many local minimums. This motivates the proposition of the PWLL objectives, which enjoy concave landscapes and have good numerical performance.

**Compliance With Llm Reviewing Policy:**

Affirmed.

**Final Justification:**

My concerns have been resolved. I keep my positive score.

**Key Questions For Authors:**

While the authors have acknowledged that the proposed PWLL objective trades accurate value estimation for good optimization properties, I am curious about whether there is any type of estimation performance description for the PWLL objective.

**Limitations:**

N.A.

**Strengths And Weaknesses:**

**Strength:**

1. This work provides reveals a new problem in the popular policy optimization literature, which is valuable and provides a principled approach for tackling it.

2. The empirical results are comprehensive and convincing.

3. The discussion is thorough and honest; limitations and connections to previous works are fairly acknowledged.

**Weakness:**

Most technical components proposed in this paper are not new. PWLL is acknowledged as reward-weighted behavioral cloning from offline RL. The optimization hardness results adapt known techniques from Mei et al. (2020) and Chen et al. (2019).
But I don't think this is a big problem as this work makes a great contribution in connecting these known optimization pathologies specifically to the large-action OPL setting and showing that the community's focus on estimation accuracy has been misplaced.

---

> ### Author Rebuttal · Authors · 2026-03-30
>
> Thank you for the positive assessment and for capturing the core contribution of the paper so well, especially your observation that *“this work makes a great contribution in connecting these known optimization pathologies specifically to the large-action OPL setting and showing that the community’s focus on estimation accuracy has been misplaced.”* This is exactly the message we aim to convey.
>
> Regarding your question on the estimation performance of PWLL, our view is that PWLL is not a value estimator. Its sole role is as a policy learning objective. Accordingly, the relevant notion of performance is not unbiasedness or consistency for value estimation, but the quality of the learned policy. We showed empirically that PWLL can have substantially worse estimation accuracy as an estimator, yet still learn stronger policies because its optimization landscape is far better behaved (Figure 6 in Appendix E.2). We will make this point more explicit in the revision.
>
> Thank you again for the detailed and encouraging review.

---

> > ### Author Rebuttal · Reviewer_qhm8 · 2026-04-04
> >
> > My concerns have been resolved. I keep my positive score.

---

> > > ### Author Response · Authors · 2026-04-07
> > >
> > > We are very glad that our rebuttal addressed your concerns, and we sincerely appreciate your positive support for the paper.

---

### Decision · Program_Chairs · 2026-04-30

**Decision:**

Accept (regular)

**Comment:**

This paper studies offline policy estimation in contextual bandits.
Prior work tackles the task by improving off-policy evaluation. This paper argues that this view neglects the role of optimization, where finding the argmax of the estimated off-policy value is prone to non-concave landscapes and local maxima.

The authors propose a novel objective that trades off the theoretical performance of the resulting policy versus the ease of optimization.

I find this a refreshing and novel perspective, however, several reviewers have strong concerns. The theoretical contribution of the paper is not overly deep, in large parts it is a summary of the existing literature. The setting is limited to contextual bandits and connections to behaviour cloning in RL are briefly mentioned but not fully elaborated.

If there is sufficient space in the conference, this would be a good submission, but it isn't a clear accept given the limited theoretical depth.